# The ESCRT protein CHMP5 promotes T cell leukemia by enabling BRD4-p300-dependent transcription

Katharine Umphred-Wilson[1,2], Shashikala Ratnayake[3,6], Qianzi Tang [4,6], Rui Wang [4,6], Sneha Ghosh Chaudhary[1], Devaiah N. Ballachanda[1], Josephine Trichka[1,2], Jan Wisniewski[1], Lan Zhou[5], Qingrong Chen[3], Daoud Meerzaman[3], Dinah S. Singer[1] & Stanley Adoro [1] ✉

Addiction to oncogene-rewired transcriptional networks is a therapeutic vulnerability in cancer cells, underscoring a need to better understand mechanisms that relay oncogene signals to the transcriptional machinery. Here, using human and mouse T cell acute lymphoblastic leukemia (T-ALL) models, we identify an essential requirement for the endosomal sorting complex required for transport protein CHMP5 in T-ALL epigenetic and transcriptional programming. CHMP5 is highly expressed in T-ALL cells where it mediates recruitment of the coactivator BRD4 and the histone acetyl transferase p300 to enhancers and super-enhancers that enable transcription of T-ALL genes. Consequently, CHMP5 depletion causes severe downregulation of critical T-ALL genes, mitigates chemoresistance and impairs T-ALL initiation by oncogenic NOTCH1 in vivo. Altogether, our findings uncover a non-oncogene dependency on CHMP5 that enables T-ALL initiation and maintenance.

Transcriptional dysregulation is a hallmark of cancer cells as oncogenes hijack the transcriptional machinery to promote expression of genes that support their survival and proliferative needs[1,2]. Studies in solid tumors and hematological cancers have demonstrated a crucial role for the bromo and extraterminal (BET) domain protein BRD4 in promoting the cancer gene program especially in MYC-dependent tumors[3–6]. BRD4 is a transcriptional coactivator with a histone acetyltransferase activity via which it mediates nucleosome decompaction, and a kinase activity that promotes RNA polymerase II (Pol II) pause release and transcription[7–9]. In BRD4-dependent cancer cells, many pro-tumorigenic genes are notably enriched for BRD4 binding at enhancers and super-enhancers marked by histone hyperacetylation marks, including histone 3 lysine 27 acetylation (H3K27ac) that is chiefly characterized by the histone acetyl transferase (HAT) p300[10,11].

Dependency of cancer gene programs on these super-enhancers distinguishes cancers from normal cells[1,2], and raises fundamental questions on how oncogenes, using the same transcription machinery as normal cells, selectively only modulate regulatory elements of cancer specific genes.

T cell acute lymphoblastic leukemia (T-ALL) is an aggressive malignancy in thymocytes[12] and a cancer with a strong BRD4 dependency[13–15]. More than half of all human T-ALL cases are caused by activating NOTCH1 mutations that constitutively generate or stabilize the intracellular NOTCH1 domain (ICN1)[16], a transcription factor that induces the proto-oncogene MYC[17]. Notably, in NOTCH1-driven T-ALL, BRD4 cooperates with ICN1 and MYC, to orchestrate a positive feedforward circuit that not only initiates and sustains the leukemogenic program[13–15] but also mediates chemoresistance in T-ALL cells[18,19].

[1]Experimental Immunology Branch, National Cancer Institute, National Institutes of Health, Bethesda, MD 20892, USA. [2]Immunology Training Program, Department of Pathology, Case Western Reserve University School of Medicine, Cleveland, OH 44106, USA. [3]Computational Genomics and Bioinformatics Branch, Center for Biomedical Informatics & Information Technology, National Cancer Institute, National Institutes of Health, Bethesda, MD 20850, USA. [4]College of Animal Science and Technology, Sichuan Agricultural University, 611130 Chengdu, China. [5]Department of Pathology and Genomic Medicine, Houston Methodist Hospital, Houston, TX 77030, USA. [6]These authors contributed equally: Shashikala Ratnayake, Qianzi Tang, Rui Wang. ✉e-mail: stanley.adoro@nih.gov

Thus, BET inhibitors which release BRD4 from chromatin down-regulate expression of T-ALL genes like *MYC* and suppress T-ALL cell survival in vitro and in pre-clinical models[5,13,19]. However, evolving resistance to BRD4 inhibitors in T-ALL and other cancers[19,20] highlights a need to better understand mechanisms of BRD4 dependency in T-ALL.

Charged multivesicular body protein-5 (CHMP5 or VPS60) is a ~35 kDa coiled-coil protein first identified in yeast[21,22] as part of the cytosolic endosomal sorting complex required for transport (ESCRT)-III machinery which promotes activation of the AAA-ATPase VPS4 to induce membrane scission[23]. Because their membrane remodeling activity is required for many cellular processes, the ESCRT machinery has long been linked to cancer, but how individual ESCRT proteins contribute to tumorigenesis is unclear and often attributed to their membrane scission activity[24,25]. Recent studies however suggest ESCRT-independent roles for CHMP5, including its function as an adaptor for deubiquitylating enzymes to stabilize client proteins[26,27]. In developing thymocytes, CHMP5 was required specifically to promote the survival TCR-signaled CD4+CD8- intermediate thymocytes survival[27]. That oncogenes hijack the same factors that promote normal thymocyte development to initiate and maintain cause T-ALL[28], raised the question whether and how CHMP5 also contributes to T cell leukemogenesis.

In this work, we show that distinct from its cytosolic localization and activity, a nuclear fraction of CHMP5 mediates the recruitment of BRD4 and the histone acetyl transferase p300 to enhancers and super-enhancers that drive transcription of T-ALL genes. Consequently, loss of CHMP5 impairs expression of critical T-ALL genes, compromising T-ALL maintenance and abrogating in vivo T-ALL initiation by activated NOTCH1.

## Results

### CHMP5 enables transcription of the T-ALL gene program

To elucidate CHMP5 function in T-ALL pathogenesis, we used short hairpins RNA (shRNA) to "knock-down" (KD) CHMP5 (Supplementary Fig. 1a) in CUTLL1 cells, a CD4+CD8+ human T-ALL caused by a t(7;9)(q34;q34) translocation of *NOTCH1* into the *TCRB* loci that results in constitutive cleavage of NOTCH1 into ICN1 by γ-secretase[29]. Although CHMP5-depleted (KD) T-ALL cells were viable similar to control (CT) cells (Supplementary Fig. 1b, c), they were severely limited in growth and displayed cell cycle arrest with more CHMP5-KD cells in S and fewer in G2/M phase (Supplementary Fig. 1d, e). Unlike cells with a defective ESCRT machinery that show impaired membrane receptor dynamics, increased cell size and DNA content[25,30], CHMP5-KD and control T-ALL cells displayed similar DNA content (Supplementary Fig. 1e), cell size (FSC) and expressed comparable levels of CD4, CD8 and TCR, the latter downregulated by anti-CD3 cross-linking antibodies similar to control cells (Supplementary Fig. 1f, g). These results suggest a critical requirement for CHMP5 in T-ALL maintenance and that, like in normal thymocytes[27], CHMP5 might be dispensable for the ESCRT machinery in T-ALL cells.

To gain insight into how CHMP5 regulated T-ALL growth, we performed RNA-seq on control and CHMP5-KD T-ALL cells. We found significant (Log$_2$ fold-change ≥ 1.2; adjusted $P < 0.05$) gene expression changes due to CHMP5 depletion, comprising 1057 upregulated and 702 downregulated genes. Enriched pathways associated with the differentially expressed genes (DEG) included interferon signaling, P53, apoptosis, all pathways with anti-leukemia activity. In contrast, CHMP5-KD cells displayed a striking downregulation of "MYC targets" (Fig. 1a, b). Given the critical role for MYC in NOTCH1-driven T-ALL[13–15], we proceeded to explore the significance of this CHMP5-MYC axis. Gene set enrichment analysis (GSEA) revealed that genes induced in T-ALL cells lacking the NOTCH-dependent *MYC* super-enhancer (N-Me)[17] were notably enriched in CHMP5-KD while those repressed upon N-Me deletion were downregulated (Supplementary Fig. 1h). Overall,

the transcriptome of CHMP5-KD T-ALL cells was the reverse of cancers with *MYC* amplifications[31] (Supplementary Fig. 1i).

*MYC* transcripts and MYC protein were severely decreased in CHMP5-KD T-ALL cells (Fig. 1c, d), in line with their downregulation of MYC target genes involved in energy metabolism, cell proliferation (Supplementary Fig. 1j). Unlike normal thymocytes in which CHMP5 promoted BCL-2 stability[27], BCL-2 proteins were unaffected by loss of CHMP5 in T-ALL (Fig. 1c), hinting at distinct functions of CHMP5 in normal thymocytes and T-ALL. MYC protein stability in CHMP5-KD T-ALL cells was similar to controls and proteasome inhibition by MG132 did not restore MYC proteins in these CHMP5-depleted cells (Supplementary Fig. 2a–c), suggesting transcriptional control of MYC expression by CHMP5. Accordingly, levels of *MYC* mRNA and MYC proteins directly correlated with CHMP5 protein and mRNA amounts in T-ALL cell lines (Fig. 1e and Supplementary Fig. 2d) and in T-ALL patients (Supplementary Fig. 2e).

To determine the specificity of CHMP5 loss on MYC expression, we transduced CHMP5-KD T-ALL cells with control ("Vector") or lentiviruses encoding murine *Chmp5* (mCHMP5) which is 99% identical in amino acid sequence to human CHMP5[32]. CHMP5 re-expression not only restored *MYC* mRNA and MYC protein expression (Fig. 1f, g) but also rescued downstream MYC-dependent processes, including mitochondria oxidation and endoplasmic reticulum biogenesis in CHMP5-depleted T-ALL cells (Supplementary Data Fig. 2f). Moreover, CHMP5 overexpression increased MYC expression in CHMP5-sufficient control (CT) cells (Fig. 1f, g), consistent with the positive correlation between CHMP5 and *MYC* amounts in T-ALL (Fig. 1e and Supplementary Fig. 2d–e). Taken together, these findings suggest a critical role for CHMP5 in enabling the T-ALL transcriptional program exemplified by high MYC expression.

### CHMP5 deficiency phenocopies MYC deficiency in T-ALL cells

MYC is dysregulated in many cancers, including T-ALL where it promotes expression of multiple genes involved in metabolism, protein synthesis and proliferation[33]. Given the dependency of MYC expression on CHMP5, we sought to define the overlap between CHMP5 loss and MYC depletion in T-ALL. While CHMP5 depletion caused MYC downregulation, MYC depletion did not disrupt CHMP5 expression (Fig. 1h, i), situating CHMP5 activity upstream of MYC in T-ALL. In vitro growth of both MYC-KD and CHMP5-KD cells was drastically compromised and accompanied by markedly reduced expression of glycolytic and oxidative phosphorylation genes including *PHB1, C1QBP, LDHA*, and *HK2* (Fig. 1j and Supplementary Fig. 2g). CHMP5 depletion, like MYC depletion, also impaired energy metabolism evidenced by significantly reduced basal and active glycolytic capacity (ECAR) and mitochondrial respiration (OCR) in T-ALL lacking these factors (Fig. 1k, l). Furthermore, both CHMP5 and MYC-depleted T-ALL cells displayed diminished ER biogenesis, mitochondria reactive oxygen species (surrogate for oxidative phosphorylation), cell size, and protein synthesis (Supplementary Fig. 2h), cellular processes regulated by MYC[33]. Thus, T-ALL cells lacking CHMP5 phenocopy MYC-depleted T-ALL cells, reinforcing downregulation of MYC as key phenotype of CHMP5 depletion in NOTCH1-activated T-ALL. We infer from these data that the severe growth defect of CHMP5-depleted T-ALL is likely driven by a combination of their downregulation of cell cycle genes and impaired energy metabolism.

### Identification of a nuclear fraction of CHMP5

The majority of ESCRT proteins, including CHMP5, are thought to be cytoplasmic[25]. However, the massive transcriptome change in CHMP5-depleted T-ALL cells raised the possibility that CHMP5 might influence transcription. To investigate this possibility, we first sought to clarify the cellular localization of CHMP5. Surprisingly, in addition to the cytosol, considerable amounts of CHMP5 proteins localized to the nucleus of human T-ALL lines (CUTLL1, MOLT-3, SUPT1) and patient-derived (PDX) T-ALL cells (Fig. 2a) as well as in normal human T cells

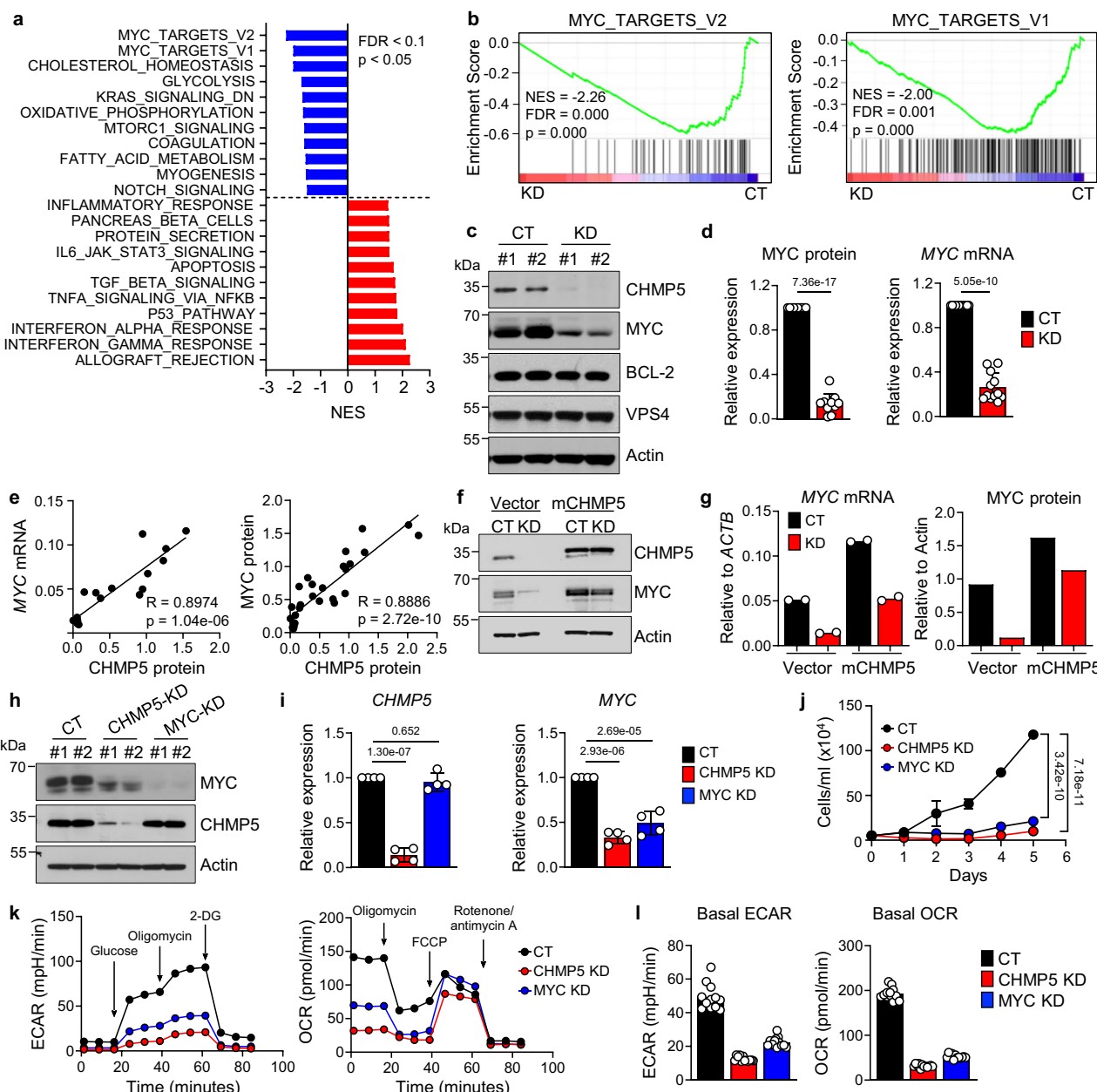

**Fig. 1 | CHMP5 promotes a T-ALL transcriptional program exemplified by MYC.**
**a** Hallmark pathway enrichment scores of differentially expressed genes (DEGs) (fold-change ≥ ±1.2; adjusted $p < 0.05$) between $n = 3$ control (CT) versus $n = 3$ CHMP5-depleted (KD) CUTLL1 cells after 5 days of selection with puromycin. $p$-value determined by Weighted Kolmogorov–Smirnov test and adjusted for multiple comparisons. FDR: false discovery rate. **b** GSEA plots of the Hallmark MYC-targets_V2 pathway (left) and MYC-targets_V1 pathway (right). $p$-value determined by Weighted Kolmogorov-Smirnov test and adjusted for multiple comparisons. NES: normalized enrichment score. $n = 3$ samples per group. **c** Western blot of the indicated proteins in CUTLL1 CT and KD whole cell lysates. Data are representative of 15 independent experiments. **d** MYC protein ($n = 10$) and mRNA ($n = 12$) expression relative to Actin and normalized to CT CUTLL1 cells. Data are mean ± SD of biological replicates pooled from 5 independent experiments. Student's t-test, two-tailed. **e** Correlation between CHMP5 protein and *MYC* mRNA (left, $n = 17$) and MYC protein (right, $n = 28$) in CUTLL1 cells. Data points are biological replicates pooled from five independent experiments. Rho (R) and $p$-values determined by Pearson Correlation test. **f** Western blot of CT and KD CUTLL1 cells transduced with

empty vector (Vector) or murine CHMP5 lentivirus (mCHMP5) for 48 h. Representative of 3 independent experiments. **g** Quantification of MYC protein and mRNA relative to Actin of cells from (**f**). qPCR data are 2 technical replicates (mRNA), and data are representative of 2 independent experiments. **h** Western blot of CUTLL1 cells transduced with 2 independent shRNAs targeting *CHMP5* or *MYC* sequences and control (CT). Representative of 4 independent experiments. **i** mRNA expression of *CHMP5* and *MYC* relative to *ACTB* and normalized to CT. Data are mean ± SD of 4 biological replicates. Statistics, one-way ANOVA, corrected for multiple comparisons using Tukey test. **j** CUTLL1 growth kinetics determined by trypan blue counting. Data are mean ± SD of cell numbers of 3 biological replicates. $p$-value determined by Two-way ANOVA. Representative of 2 independent experiments. **k** Extracellular acidification rate (ECAR, left), and oxygen consumption rate (OCR, right) kinetics. Data are presented as mean of 6 technical replicates from 2 biological replicates per group ($n = 12$). 2-DG, 2-deoxyglucose; FCCP, carbonyl cyanide-4-(trifluoromethoxy) phenylhydrazone. **l** Basal ECAR and OCR values of indicated CUTLL1 cells from data in (**k**). Data points are 6 technical replicates from 2 biological replicates ($n = 12$). Source data are provided as a Source Data file.

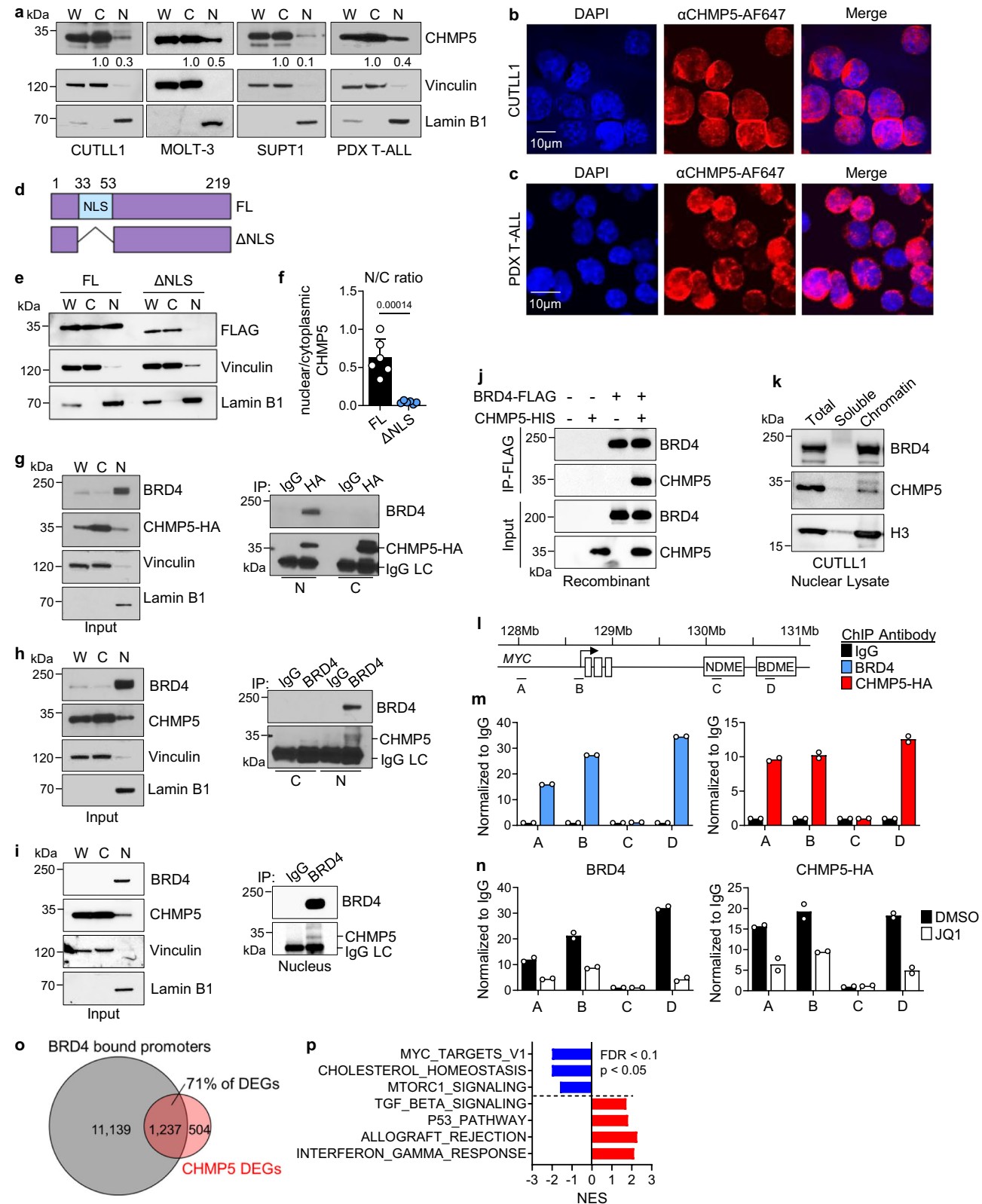

(Supplementary Fig. 3a). Confocal microscopy revealed CHMP5 protein signals throughout the nucleus of T-ALL cells that were diminished in CHMP5-KD controls (Fig. 2b, c and Supplementary Fig. 3b). Overall, nuclear CHMP5 comprised ~10–50% of cytosolic CHMP5 levels (Fig. 2a and Supplementary Fig. 3a).

To elucidate the basis of its nuclear localization, we examined the amino acid sequence of CHMP5 across various species for a nuclear localization signal (NLS). Interestingly, beginning in jawed vertebrates (including mice and humans), the N-terminal amino acid sequence of CHMP5 became highly conserved and contained a putative bipartite NLS[34] that was absent in invertebrate eukaryotes (Supplementary Fig. 3c, d). To functionally validate this putative NLS, we deleted the corresponding amino acid residues (Fig. 2d) and assessed the localization of NLS-deficient (ΔNLS) CHMP5 proteins. While, like full-length

**Fig. 2 | Identification of nuclear CHMP5-BRD4 interaction on chromatin.**
**a** Western blot of fractionated CUTLL1, MOLT-3, SUPT1, and patient derived
xenograft (PDX) T-ALL cells. W, whole cell; C, cytoplasmic; and N, nuclear lysates.
Nuclear CHMP5 band intensities relative to the cytoplasmic band are indicated.
Representative of 4 independent experiments. **b, c** Immunofluorescence images of
CUTLL1 (**b**) and T-ALL PDX (**c**) stained with anti-CHMP5 primary antibody followed
by a secondary AF647 antibody (Red) and DAPI for nuclear staining (blue). Images
are representative of 6 samples (3 CUTLL1, 3 PDX) from 3 independent experi-
ments. Scale bars indicate 10 μm. **d** Schematic representation of the full length (FL)
and NLS truncation mutant (ΔNLS) constructs of murine CHMP5. Created in
BioRender. Umphred-Wilson, K. (2025) https://BioRender.com/dzfs72l. **e** Western
blot of fractionated CUTLL1 cells transduced with FLAG-tagged full-length CHMP5
(FL) or ΔNLS mutant encoding lentivirus. Representative of 6 experiments.
**f** Quantification of the nuclear/cytoplasmic ratio of FL-CHMP5 and ΔNLS-CHMP5
determined by densitometry. Data points are biological replicates from $n = 6$
independent experiments. Student's $t$ test, two-tailed. **g** Western blot of fractio-
nated CUTLL1 cells transduced with CHMP5-HA subjected to immunoprecipitation
with isotype (IgG) or anti-HA antibodies. Representative of 3 experiments.
**h** Western blot of fractionated CUTLL1 cells subjected to immunoprecipitation with
IgG or anti-BRD4 antibodies. Representative of 3 experiments. **i** Western blot of

fractionated PDX T-ALL cells subjected to immunoprecipitation with IgG or anti-
BRD4 antibodies. Representative of 2 experiments. **j** Anti-FLAG immunoprecipita-
tion of BRD4-FLAG and CHMP5-His recombinant proteins. Representative of 2
experiments. **k** Nuclear fractionation of CUTLL1 cells into soluble and chromatin-
bound fractions. Representative of 2 independent experiments. **l** Schematic of the
*MYC* gene locus indicating ChIP-qPCR primer binding: A, enhancer; B, promoter; C,
NDME (NOTCH-dependent *MYC* enhancer); D, BDME (BRD4-dependent *MYC*
enhancer). **m** ChIP-qPCR of anti-BRD4 and CHMP5-HA at the *MYC* locus normalized
to isotype (IgG). Data are 2 technical replicates, representative of 3 independent
experiments. **n** ChIP-qPCR of anti-BRD4 and CHMP5-HA normalized to IgG from
CUTLL1 cells treated with vehicle (DMSO) or 500 nM of JQ1 for 18 h. Data are 2
technical replicates, representative of 2 independent experiments. **o** Venn-diagram
of BRD4 bound genes (determined by ChIP-seq GSE51800) and DEGs between CT
and KD CUTLL1 cells (Fig. 1a). The BRD4 ChIP-seq dataset are available in the NCBI
Gene Expression Omnibus database under accession code GSE51800[41]. **p** Pathway
analysis of the overlapping BRD4-bound genes and DEGs between CT and KD
CUTLL1 cells (CT, $n = 3$; KD, $n = 3$ samples each) in (**o**). *p*-value determined by
Weighted Kolmogorov-Smirnov test and adjusted for multiple comparisons.
Source data are provided as a Source Data file.

CHMP5 (FL), ΔNLS-CHMP5 proteins still interacted with VTA1/Lip5, an
ESCRT partner of CHMP5[35] (Supplementary Fig. 3e), it mainly only
localized to the cytosol (Fig. 2e, f). This suggests that nuclear locali-
zation of CHMP5 in T-ALL cells was at least in part mediated by an NLS-
dependent import mechanism.

### Nuclear CHMP5 interacts with BRD4 on chromatin

The nuclear localization of CHMP5 prompted us to hypothesize that it
might function as part of the nuclear machinery regulating transcrip-
tion of T-ALL genes. Since in NOTCH1-driven T-ALL, ICN1 induces
transcription of pro-leukemogenic genes in cooperation with BRD4
and MYC[13–15], we investigated whether nuclear CHMP5 interacted with
these transcription factors. Due to lack of suitable CHMP5 immuno-
precipitation antibodies, we transduced T-ALL cells with hemaggluti-
nin (HA)-tagged CHMP5 and performed anti-HA immunoprecipitation
from these cells. Notably, while CHMP5 did not interact with ICN1 or
MYC (Supplementary Fig. 3f), it co-immunoprecipitated with endo-
genous BRD4 in nuclear T-ALL lysates (Fig. 2g and Supplementary
Fig. 3g). By reverse immunoprecipitation of endogenous BRD4, we
confirmed the CHMP5-BRD4 interaction in nuclear lysates from T-ALL
cell line and primary PDX human T-ALL (Fig. 2h, i) as well as in HEK293T
cells co-transfected with plasmids encoding these proteins (Supple-
mentary Fig. 3h). In cell-free conditions, recombinant BRD4 "pulled-
down" CHMP5 (Fig. 2j). Moreover, like BRD4, nuclear CHMP5 was
present in the chromatin fraction of the nucleus (Fig. 2k). These data
support a direct interaction between CHMP5 and BRD4 in the nucleus.

To determine, if as suggested by their interaction, CHMP5 bound
chromatin sites as BRD4, we performed anti-BRD4 and anti-HA
(CHMP5) chromatin immunoprecipitation-qPCR (ChIP-qPCR) on
T-ALL cells transduced with HA-tagged CHMP5. Both antibodies
pulled-down chromatin relative to isotype antibody (Supplementary
Fig. 3i), validating our anti-HA ChIP approach. Focusing on the *MYC*
locus (Fig. 2l), ChIP-qPCR revealed CHMP5 binding on chromatin at
sites that overlapped with BRD4 binding, including at the *MYC*
enhancer, promoter, and a BRD4-dependent super-enhancer
("BDME")[18] (Fig. 2m). CHMP5 was not enriched at the ICN1-specific
("NDME") super-enhancer which is not bound by BRD4[18] (Fig. 2m),
indicating that CHMP5 bound chromatin at regulatory DNA elements
via its association with BRD4. Accordingly, treatment with the BET
inhibitor JQ1 which releases BRD4 from chromatin[36] diminished
CHMP5 binding across the *MYC* locus (Fig. 2n). Thus, nuclear CHMP5 is
associated with chromatin at least in part through its interaction with
BRD4, including at regulatory DNA elements that control MYC
expression and suggest that CHMP5 might modulate BRD4-dependent

transcription. In support of this possibility, majority (~71%) of DEGs in
CHMP5-KD T-ALL cells have promoters bound by BRD4 and were
genes represented in the topmost perturbed pathways in CHMP5-
depleted T-ALL cells (Fig. 2o, p).

### CHMP5 promotes Pol II release at BRD4-dependent T-ALL genes

BRD4 promotes transcriptional elongation in part by activating
promoter-proximal Pol II pause release[8,9,37,38]. We thus asked whether
CHMP5 modulated BRD4-dependent regulation of Pol II pause
release by performing genome-wide ChIP-seq for Pol II and BRD4 in
CHMP5-KD or control (CT) T-ALL cells. ChIP-seq analysis showed a
modest increase in BRD4 binding at transcriptional start sites (TSS)
but largely unchanged BRD4 binding across gene bodies and tran-
scriptional end sites (TES) in CHMP5-KD cells (Supplementary
Fig. 4a, b). However, these CHMP5-KD T-ALL cells displayed
decreased Pol II occupancy at the TES with a corresponding increase
of Pol II at promoters (Fig. 3a–c and Supplementary Fig. 4c), sug-
gesting impaired Pol II pause-release. Since BRD4 travels with Pol II
during transcription[39], the increase in BRD4 occupancy at promoters
may reflect its binding to stalled Pol II. Importantly, diminished Pol II
at the TES correlated with gene expression, as reduction in Pol II
density at the TES was more significant in downregulated DEGs
compared to upregulated DEGs in CHMP5-KD T-ALL cells (Supple-
mentary Fig. 4d).

To quantify the extent to which CHMP5 loss impacted Pol II
pause release, we computed genome-wide Pol II traveling ratios
(TR), the relative ratio of Pol II density at promoter-proximal
regions relative to its density across the gene body (Fig. 3d)[9,40].
We found higher TRs for BRD4-target genes[41] in CHMP5-KD T-ALL
cells (Fig. 3e), corroborating impaired Pol II pause release at
promoters and decreased transcription of these T-ALL genes.
MYC target genes, which were the most downregulated in
CHMP5-KD T-ALL, had especially increased TRs (Fig. 3f), likely
also reflecting impairment in the Pol II release activity of MYC[42].
Genome-wide TR correlation show increased TRs for critical
T-ALL genes including *MYC*, *XBP1*, *TCF7*, *PHB2*, and *CDK7*
in CHMP5-KD T-ALL cells (Fig. 3g) and Pol II density at the 3'-ends
of these genes was strikingly reduced (Fig. 3h). Validating
their dependency on BRD4, treatment with JQ1 downregulated
expression of genes (e.g., *MYC*, *XBP1* and *TCF7*) with higher
TRs in CHMP5-KD T-ALL cells was downregulated by JQ1
and this downregulation was comparable to the effect of
CHMP5 depletion (Fig. 3i). That JQ1 further diminished *MYC*
transcript expression (but not *XBP1* or *TCF7*) in CHMP5-KD

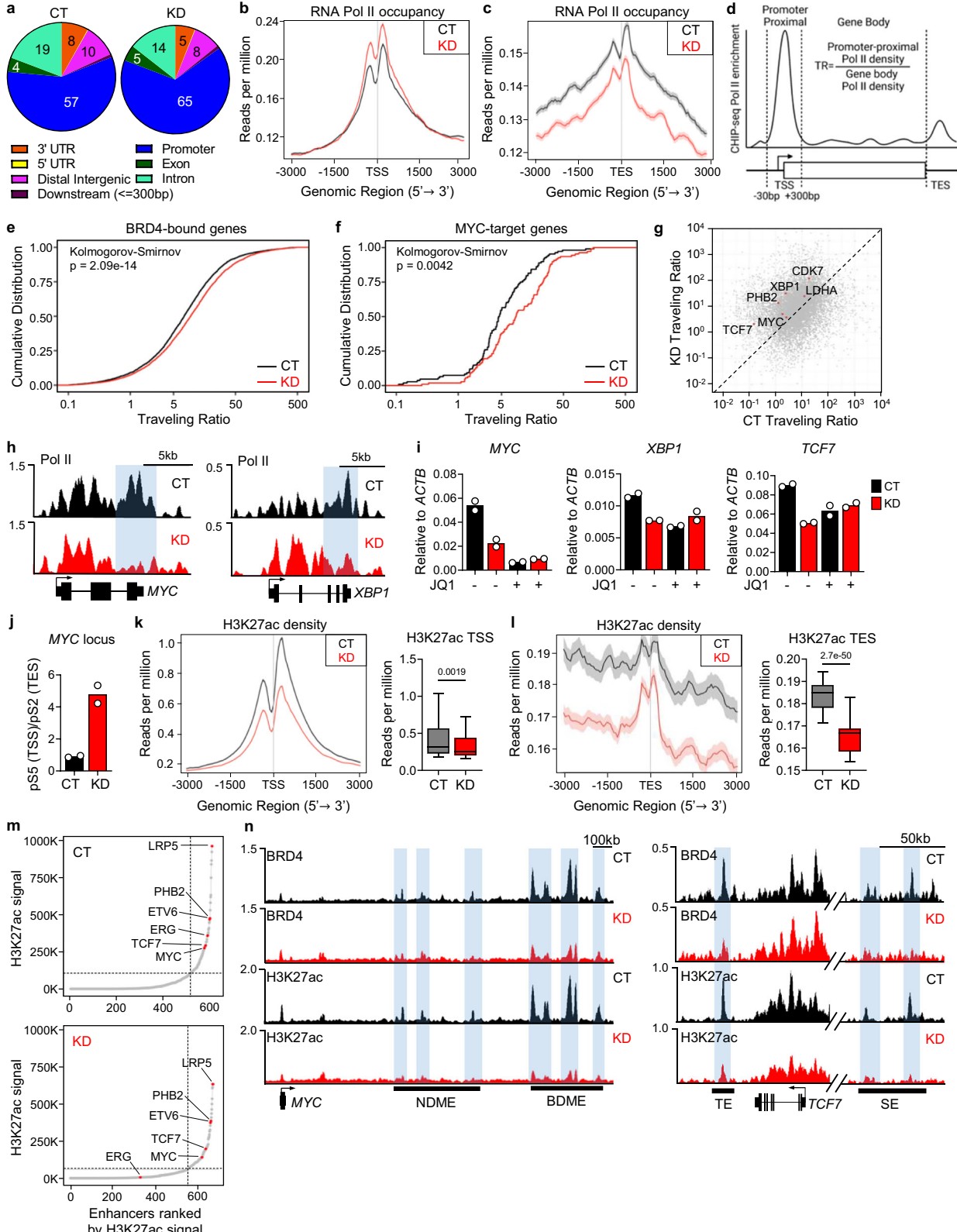

cells (Fig. 3i) imply that CHMP5-independent mechanisms also promote BRD4-dependent regulation of *MYC*.

Phosphorylation of the C-terminal domain (CTD) of the large Pol II subunit dictates stages of transcription, with phosphorylation on Serine 5 (pS5) during initiation, and Serine 2 (pS2) during elongation[43]. Using ChIP-qPCR, we detected higher pS5-Pol II at the TSS and decreased pS2-Pol II at the TES (Supplementary Fig. 4e), which resulted

in a higher pS5(TSS)/pS2 (TES) ratio (Fig. 3j) at the *MYC* gene in CHMP5-KD T-ALL cells. These results corroborate that CHMP5 promotes Pol II release at BRD4-dependent genes like *MYC*, in line with Pol II stalling and overall increased TR of BRD4-dependent genes in these cells. Importantly, this effect of CHMP5 was likely dependent on its nuclear translocation, as ΔNLS-CHMP5 proteins failed to rescue MYC or XBP1 expression like the full-length CHMP5 protein (Supplementary Fig. 4f),

**Fig. 3 | CHMP5 mediates BRD4-driven Pol II pause release and super enhancer formation. a** Pie chart of genome-wide Pol II binding in control (CT) and CHMP5-depleted (KD) CUTLL1 cells determined by ChIP-seq. **b** Metaplot of Pol II density at transcriptional start site (TSS) of all genes from (**a**). **c** Metaplot of Pol II density at the transcriptional end site (TES) of all genes from (**a**). Data in (**a**–**c**) are genome-wide Pol II signals and are representative of 2 biological replicates; shaded area along line graphs represent standard error. **d** Schematic for the calculation of Pol II traveling ratio. **e, f** Pol II traveling ratio for BRD4-bound (n = 4138 genes) (**e**) and the Hallmark MYC-Target genes (n = 118) from GSEA in Fig. 1a, b (**f**). Kolmogorov–Smirnov test is two-sided and not adjusted for multiple comparisons. **g** Dot plot comparison of Pol II traveling ratios between CT and KD cells highlighting BRD4 and MYC-target genes. **h** Pol II binding tracks in CT and KD CUTLL1 cells at the *MYC* and *XBP1* gene loci. 3′-end of genes are highlighted. **i** mRNA expression of *MYC, XBP1*, and *TCF7* in CUTLL1 CT and KD cells treated with DMSO (−) or JQ1 (+) for 18 h. Data are 2 technical replicates, representative of 3 experiments. **j** The ratio of pS5 at the TSS to pS2 at the TES at the *MYC* locus in CT and KD cells based on the ChIP qPCR in (Supplementary Fig. 4e). Data points are 2 technical replicates. **k, l** Metaplot and boxplot of H3K27ac density at TSS (**k**) and TES (**l**) of active genes determined by ChIP-seq. Data in (**k, l**) are calculated for H3K27ac signals at active genes and representative of 2 biological replicates; shaded area along line graphs represent standard error; boxplot limits are the 25th and 75th percentiles, with center line indicating the median. Whiskers extend to the minimum and maximum values. n = 101, Student's *t* test, two-tailed. **m** Hockey stick plots of ranked genome-wide H3K27ac signals in CT (top) and KD (bottom) CUTLL1 cells. Positions of key T-ALL genes are highlighted. **n** BRD4 and H3K27ac ChIP-seq tracks at the *MYC* and *TCF7* gene loci in CUTLL1 cells. BDME BRD4-dependent MYC enhancer, NDME NOTCH-dependent MYC enhancer, SE super-enhancer, TE typical enhancer. Source data are provided as a Source Data file.

despite preserved interaction with the ESCRT protein VTA1 (Supplementary Fig. 3e).

## CHMP5 loss disrupts H3K27 acetylation and super-enhancers

Depletion of CHMP5 did not impact promoter-proximal (TSS) BRD4 binding, which remained largely unchanged in CHMP5-KD T-ALL cells (Supplementary Fig. 4a, b), so we assessed whether it instead controlled BRD4 binding and/or activity at distal enhancers and super-enhancers marked by hyperacetylated histone modifications, including H3K27ac[44,45]. To this end, we performed global H3K27ac specific ChIP-seq and found significantly diminished H3K27ac density at the TSS, gene bodies, and TES of active genes in CHMP5-KD T-ALL cells (Fig. 3k, l and Supplementary Fig. 4g). Accordingly, in the absence of CHMP5, most actively transcribed (identified by positive H3K27ac signal) T-ALL genes showed impaired Pol II traveling (Supplementary Fig. 4h).

We then defined enhancers and super-enhancers by ranking genome-wide H3K27ac signal intensity which reveals asymmetrically distributed and markedly enriched H3K27ac signals at super-enhancers[44,45]. While in control T-ALL cells (CT), "hockey stick" plots of H3K27ac signals showed disproportionately higher H3K27ac density at super-enhancers of genes like *MYC, TCF, ERG*, and *ETV6* as previously reported[38], CHMP5 depletion markedly diminished these H3K27ac enhancers and super-enhancers (Fig. 3m), in line with diminished expression of these genes in CHMP5-KD T-ALL cells (Supplementary Fig. 4i). Correspondingly, we noted severely reduced H3K27ac and BRD4 occupancy at enhancers and super-enhancers of these T-ALL genes (Fig. 3n and Supplementary Fig. 4j). H3K27ac of the NOTCH-dependent *MYC* enhancer (NDME)[18] was only mildly affected by loss of CHMP5 (Fig. 3n), likely reflecting that CHMP5 did not interact with ICN1 (Supplementary Fig. 3f) and may not regulate this enhancer in T-ALL cells.

These results document that CHMP5 regulates the epigenetic machinery which mediates H3K27ac of enhancers and super-enhancers, sites of BRD4 binding. Accordingly, BRD4 occupancy at these super-enhancers was also diminished in CHMP5-depleted T-ALL cells (Supplementary Fig. 4k). Overall, loss of CHMP5 appears to impair BRD4 occupancy mainly at enhancers and super-enhancers (but not at promoters/TSS), suggesting that CHMP5 regulates epigenetic events selectively at distal regulatory elements. Furthermore, consistent with the lack of synergy between JQ1 and CHMP5 depletion in suppressing T-ALL cell viability (Supplementary Fig. 4l), our data positions CHMP5 upstream of BRD4 recruitment to H3K27ac marked chromatin.

## Potentiation of BRD4-p300 interaction by CHMP5

In normal and cancer cells, the HAT enzyme p300 mediates hyperacetylation of enhancer and super-enhancer to which BRD4 is recruited[10,11]. Chromatin-bound BRD4 recruits and augments p300 HAT activity, a positive feed-forward mechanism that further enriches hyperacetylation of enhancers and super-enhancers[46–48]. Thus, to determine how CHMP5 promoted H3K27ac at gene enhancer and super-enhancers, we evaluated the effect of CHMP5 depletion on the BRD4-p300 interaction. While BRD4 interaction with components of the transcriptional machinery (e.g., Pol II, MED1) were intact, its interaction with p300 was notably reduced in the absence of CHMP5, despite an overall increase in p300 proteins in CHMP5-KD cells (Fig. 4a, b and Supplementary Fig. 5a, b). This increase in p300 proteins in CHMP5-KD T-ALL cells is likely due to its impaired interaction with BRD4 as targeted degradation of BRD4 using the PROTAC degrader MZ1[49] also resulted in increased p300 proteins in CHMP5-sufficient cells (Supplementary Fig. 5c, d).

To directly assess the impact of CHMP5 on the BRD4-p300 interaction, we next quantified recombinant BRD4 and p300 binding in the presence and absence of recombinant CHMP5 protein and found that more p300 proteins co-immunoprecipitated with BRD4 when CHMP5 was present (Fig. 4c). Immunoprecipitation in HA-CHMP5-expressing T-ALL cell nuclear lysates revealed p300 binding to CHMP5 (Supplementary Fig. 5e) and recombinant CHMP5 immunoprecipitated with p300 in cell-free assays (Fig. 4d), supporting a direct interaction between both proteins. Since CHMP5 also binds BRD4 (Fig. 3g–j), these findings suggest that CHMP5 promotes H3K27ac by augmenting the BRD4-p300 interaction at enhancers and super-enhancers which in turn assemble co-factors that shape chromatin architecture to induce promoter bound Pol II activity (Fig. 4e).

We next examined the consequence of CHMP5 depletion on p300 chromatin occupancy, focusing on the *MYC* locus. Loss of CHMP5 reduced p300 occupancy at the *MYC* enhancer and super-enhancer (BDME), but not at the promoter (Fig. 4f). To understand whether CHMP5's effect on p300 binding and activity on chromatin was dependent on BRD4 by T-ALL treating cells with JQ1. Notably, unlike BRD4 binding at the *MYC* super-enhancer (BDME) which was reduced by JQ1 in control and CHMP5-KD cells, p300 and H3K27ac were diminished at the *MYC* BDME by JQ1 similar to the effect of CHMP5 depletion (Fig. 4g), consistent with a model in which CHMP5 specifically mediates p300-BRD4 interaction and H3K27ac.

To further validate the mechanistic link between CHMP5 and p300, we examined how loss of CHMP5 compared with p300 disruption using small molecule inhibitors. Both CCS1477, which disrupts p300 binding to chromatin[50], and A485, which inhibits p300 HAT activity[51], reduced *MYC* expression with the same potency as CHMP5 depletion (Fig. 4h). *MYC* downregulation in T-ALL cells by these p300 inhibitors was associated with diminished H3K27ac as well as decreased BRD4 and p300 occupancy at the BDME similar to the impact of CHMP5 depletion (Fig. 4i). The analogous effect of CHMP5 depletion and p300 inhibition on *MYC* transcription combined with the lack of synergy between p300 inhibition (by CCS1477) and CHMP5 loss on T-ALL viability (Supplementary Fig. 5f) further corroborate that CHMP5 functions upstream of p300 (and BRD4).

Collectively, our data suggest a model in which CHMP5 is indispensable for p300-catalyzed acetylation (e.g., H3K27ac) of the super-enhancer that drive expression T-ALL genes (Fig. 4e and

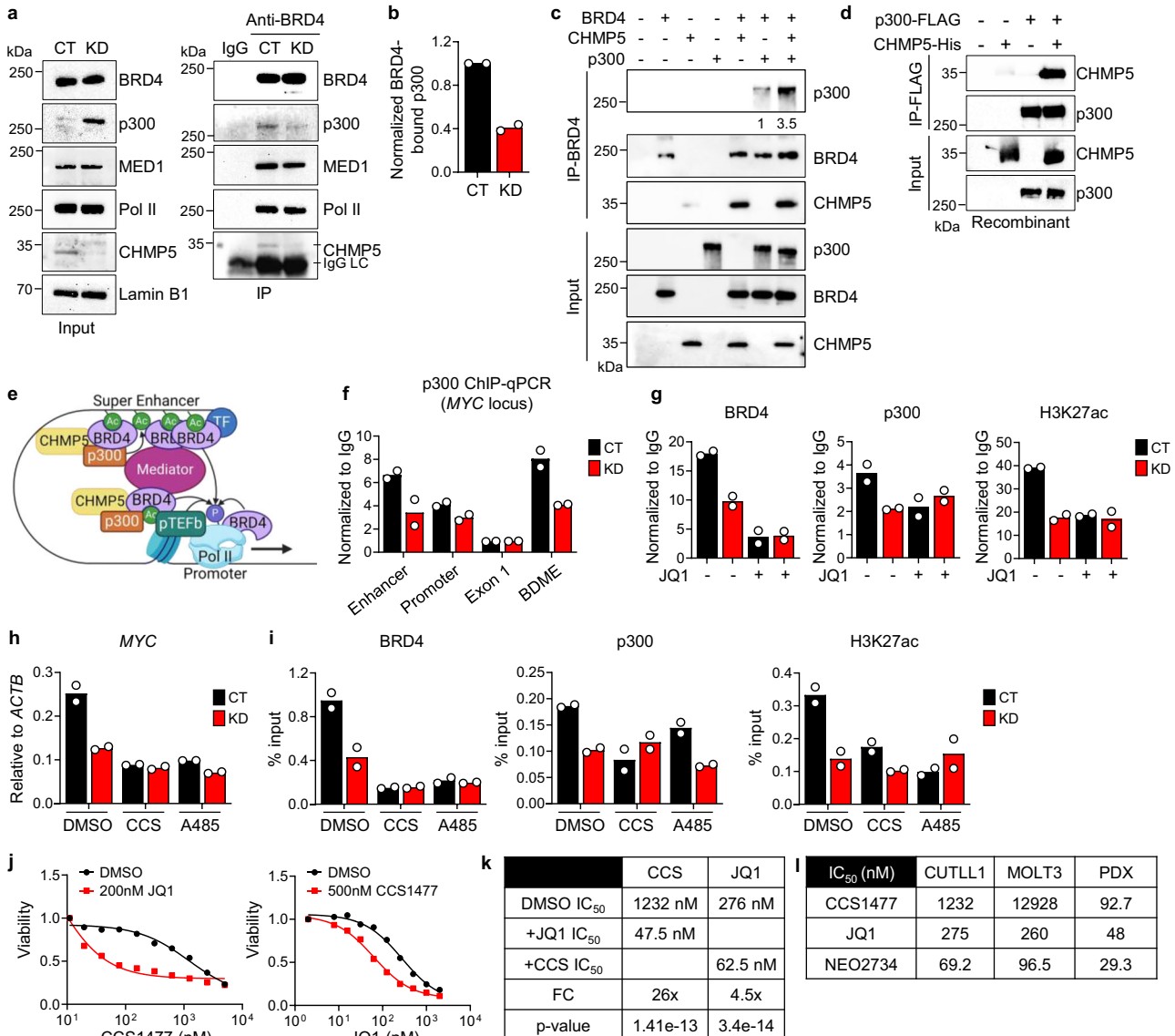

**Fig. 4 | CHMP5 promotes the interaction between BRD4 and p300.**
**a** Immunoprecipitation with isotype (IgG) or anti-BRD4 antibody from CT and KD CUTLL1 nuclear lysate. Representative of 2 independent experiments. **b** Quantification of p300 bound to BRD4 from (**a**) calculated as p300-to-BRD4 ratio and normalized to control (CT) cells. Data are two biological replicates from independent experiments, normalized to control. **c** Recombinant BRD4, p300, and CHMP5 immunoprecipitation with anti-BRD4 antibody. Ratio of p300 bound to BRD4 (quantified by densitometry) relative to no-CHMP5 lane 6 is indicated. Data is representative of 2 independent experiments. **d** Recombinant p300-FLAG and CHMP5-His immunoprecipitation with anti-FLAG antibody. Representative of 2 experiments. **e** The predicted model of how CHMP5 mediates the p300-BRD4 interaction to promote histone acetylation and productive transcription, Created in BioRender. Umphred-Wilson, K. (2025) https://BioRender.com/w4989rg. **f** ChIP-qPCR of p300 at the MYC locus normalized to IgG in CT and KD CUTLL1 cells. Data

are 2 biological replicates from independent experiments. **g** ChIP-qPCR of BRD4, p300, and H3K27ac at the *MYC* BDME in CUTLL1 cells treated with DMSO (−) or 500 nM of JQ1 (+) for 18 h. Data are 2 technical replicates, representative of 3 experiments. **h** RT-qPCR of *MYC* in CT and KD cells treated with 500 nM CCS1477 or 2 μM A485 for 18 h. Data are 2 technical replicates, representative of 2 experiments. **i** BRD4, p300, and H3K27ac ChIP-qPCR on the MYC BDME from the cells in (**h**). Data are 2 technical replicates, representative of 2 experiments. **j** Viability of parental CUTLL1 cells treated with CCS1477 or JQ1 plus 200 nM JQ1 (left) or 500 nM CCS1477 (right) for 72 h. Data are mean of 3 technical replicates, representative of 2 experiments. **k** IC$_{50}$ for CUTLL1 cell killing by CCS1477 and JQ1 combinations. IC$_{50}$ calculated by non-linear best-fit analysis. p-value calculated by 2-way ANOVA. FC, fold-change of DMSO vs +JQ1 or +CCS. **l** IC$_{50}$ of MOLT-3, CUTLL1, and T-ALL PDX treated with CCS1477, JQ1, or NEO2734. Source data are provided as a Source Data file.

Supplementary Fig. 5g). In CHMP5-sufficient T-ALL, BRD4 is recruited to these super-enhancers where it recruits p300 that catalyzes addition H3K27ac, further amplifying BRD4-p300 recruitment to super-enhancers. Long-range interaction of super-enhancers with promoter-proximal factors induce transcriptional elongation[46–48]. In CHMP5-depleted T-ALL cells by contrast, super-enhancer regions are hypoacetylated, diminishing their p300-BRD4 recruitment, and impairing transcriptional elongation (Supplementary Fig. 5g). By highlighting the

dual importance of BRD4 and p300 in T-ALL gene transcription, this model predicts that combined inhibition of p300 and BRD4 should be superior to inhibition of BRD4 or p300 alone in suppressing T-ALL. Indeed, co-inhibition of BRD4 (JQ1) and p300 (CCS1477) significantly improved the efficacy of either inhibitor alone (Fig. 4j, k). Furthermore, we document that a dual p300/BRD4 inhibitor NEO2734[52,53] suppressed T-ALL cell growth with a 3–10-fold lower IC$_{50}$ compared to JQ1 or CCS1477 alone (Fig. 4l).

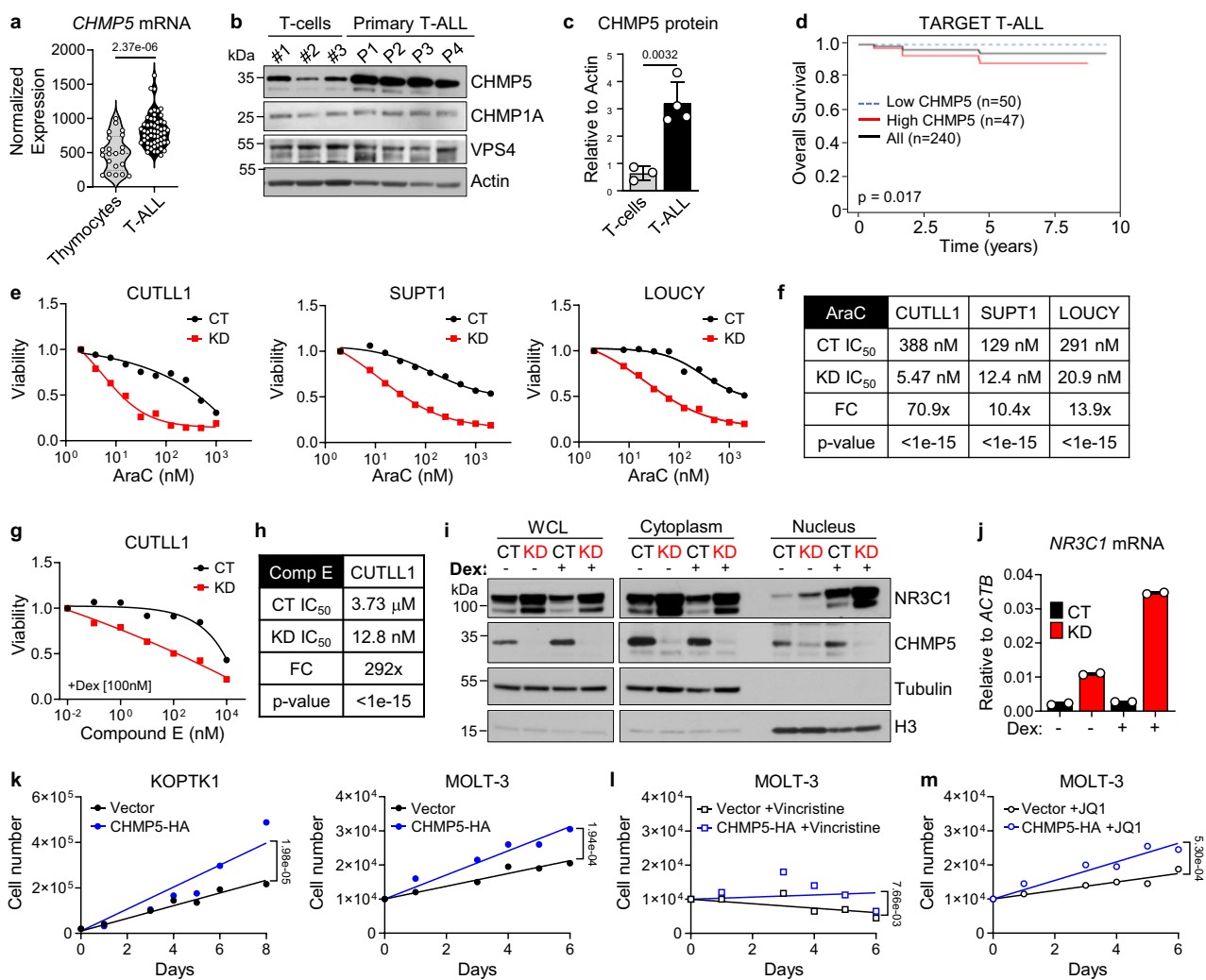

**Fig. 5 | High CHMP5 expression correlates with poor prognosis and promotes chemoresistance in T-ALL. a** *CHMP5* mRNA expression in normal thymocytes (*n* = 21) and primary T-ALL (*n* = 57) samples (GSE33470, GSE33469). Student's *t* test, two-tailed. The *CHMP5* publicly available data used in this study are available in the NCBI Gene Expression Omnibus database under accession code GSE33470[55] and GSE33469[54]. **b** Western blot of indicated proteins in T cells from healthy donors (*n* = 3) and primary human T-ALL samples (*n* = 4). Representative of 3 experiments. **c** Quantification of CHMP5 protein relative to Actin from (**b**). Data presented as mean ± SD of biological replicates (*n* = 3 T cells, *n* = 4 T-ALL). Student's *t* test, two-tailed. **d** Overall survival of pediatric T-ALL patients (TARGET T-ALL) expressing high (top 20%) and low (bottom 20%) levels of CHMP5. *p* = 0.017, Log-rank test. Low CHMP5 *n* = 50, High CHMP5 *n* = 47, All patients *n* = 240. The TARGET-TALL publicly available data used in this study are available in the NCBI database of Genotypes and Phenotypes under accession code phs000464.v7.p1[56]. **e** Viability of control (CT) and CHMP5-depleted (KD) CUTLL1, SUPT1, LOUCY cells treated with cytarabine (AraC) for 3 days. Data are presented as mean of 3 technical replicates and representative of 2 independent experiments. **f** IC50 for AraC in CUTLL1, SUPT1, and LOUCY cells from (**e**). IC50 calculated by non-linear best-fit analysis. *p*-value

calculated by 2-way ANOVA. FC, fold-change in CT versus KD IC50. **g** Viability of CT and KD CUTLL1 cells treated with Compound E (Comp E) plus 100 nM dexamethasone (Dex) for 3 days. Data are presented as mean of 3 technical replicates. **h** IC50 for Compound E in CT and KD CUTLL1 cells. IC50 calculated by non-linear best-fit analysis. p-value calculated by 2-way ANOVA. FC, fold-change in CT versus KD IC50. **i, j** Western blot of fractionated CUTLL1 cells treated with vehicle (DMSO) or 1 μM dexamethasone (Dex) for 18 h and expression of *NR3C1* in CUTLL1 cells from (**j**). qPCR data are 2 technical replicates and results are representative of 3 independent experiments. **k** KOPTK1 and MOLT-3 cells were transduced with vector or mCHMP5 lentivirus also encoding GFP. After 48 h, GFP+ cells were sorted and cultured for 6-8 days with trypan blue counting. **l, m** Proliferation of control or mCHMP5-overexpressing MOLT-3 cells. Sorted GFP+ cells (as in **k**) were cultured with 100 nM JQ1 (**l**) or 1 nM Vincristine (**m**) for 6 days. Each data point in (**k–m**) are average cell numbers of 2 biological replicates. Cell growth curves were generated with linear regression analysis and *p*-statistics of differences between the linear regression slopes of the curves are indicated. Data are representative of 2 independent experiments. Source data are provided as a Source Data file.

## High CHMP5 expression predicts poor T-ALL prognosis

We next sought to understand the clinical significance of CHMP5 to human T-ALL pathogenesis. We found that among cancers, lymphoid leukemias expressed the highest amount of CHMP5 protein (Supplementary Fig. 6a). More specifically, comparison of CHMP5 expression in pediatric T-ALL samples versus normal thymocytes (GSE33470, GSE33469)[54,55] revealed higher *CHMP5* transcripts in T-ALL cells (Fig. 5a) which translated to >5-fold more CHMP5 proteins in primary human T-ALL relative to healthy T-cells (Fig. 5b, c). Expression of other ESCRT proteins like *CHMP1A* and *VPS4A* was comparable between

normal T cells and T-ALL cells (Fig. 5b), in line with a unique role for CHMP5 in T-ALL pathogenesis. Higher CHMP5 expression was also characteristic of a panel of subtypes of human T-ALL cell lines (Supplementary Fig. 6b, c).

To determine whether CHMP5 expression levels had any prognostic significance in human T-ALL, we evaluated the correlation between *CHMP5* expression and the survival of patients in the pediatric TARGET T-ALL cohort (dbGaP phs000464)[56]. While overall survival was high in this cohort, patients with the highest (top 20%) *CHMP5* expression had worse overall survival (Log-rank test, *P* = 0.017)

compared to T-ALL patients with the lowest (bottom 20%) *CHMP5* expression levels (Fig. 5d). Importantly, expression of *VPS4A* and *CHMP1A* transcripts did not correlate with T-ALL patient survival (Supplementary Fig. 6d, e), reinforcing a unique role for CHMP5 in T-ALL pathogenesis. Furthermore, *CHMP5* expression was significantly higher in T-ALL cells from adult patients that did not achieve complete remission[57] (Supplementary Fig. 6f). Thus, high expression of CHMP5 distinguishes T-ALL from normal T cells and levels of *CHMP5* in T-ALL cells can be a prognostic indicator of T-ALL patient survival outcomes.

### CHMP5 mediates chemoresistance-mechanisms in T-ALL

The poor overall survival of T-ALL patients with the highest expression of CHMP5 suggests that CHMP5 might contribute to patient T-ALL response to treatment and prompted us to investigate how CHMP5-deficiency impacted whether T-ALL cell response to chemotherapy. To this end, we next assessed the impact of CHMP5 depletion on T-ALL cell sensitivity to cytarabine (AraC), a drug included as part of a T-ALL inductive treatment regimen[58,59]. CHMP5-depletion markedly increased AraC-induced cell death in both NOTCH1-dependent (CUTLL1 and SUPT1) and NOTCH1-independent (LOUCY) T-ALL cells, which reflected in a >10-fold reduction in the $IC_{50}$ of AraC in CHMP5-KD T-ALL (Fig. 5e, f).

We evaluated CHMP5-depleted T-ALL response to the targeted therapy combination of γ-secretase inhibitors plus dexamethasone ("GSI+Dex"), in trial for NOTCH1-dependent T-ALL[60–62]. Unlike control cells that are resistant to this treatment[62], CHMP5-depleted CUTLL1 T-ALL cells were potently inhibited by GSI+Dex (Fig. 5g, h). Notably, increased sensitivity to GSI+Dex was associated with higher expression and nuclear translocation of the glucocorticoid receptor NR3C1, which upregulated NR3C1-target genes *NR3C1* and *BIM* in CHMP5-KD T-ALL cells (Fig. 5i, j and Supplementary Fig. 6g). Since NR3C1 is repressed by HES1 downstream of activated NOTCH1[62], upregulation of NR3C1 in CHMP5-KD CUTLL1 cells (Fig. 5i, j) could be due to their decreased expression of *HES1* (Supplementary Fig. 6h, i). Of note, regulation of NR3C1 and T-ALL response to GSI+Dex by CHMP5 was specific to NOTCH1-dependent T-ALL (CUTLL1) as NOTCH1-independent LOUCY T-ALL cells were unresponsive to GSI+Dex (Supplementary Fig. 6j–l).

We then explored the consequence of CHMP5 overexpression on T-ALL chemoresistance using KOPTK1 and MOLT-3 human T-ALL cells, whose CHMP5 levels were lower than other human T-ALL lines. At baseline, CHMP5 overexpression enhanced the growth of KOPTK1 and MOLT-3 cells (Fig. 5k). Importantly, overexpression of CHMP5 not only mitigated the efficacy of the chemotherapy drug vincristine (Fig. 5l), but it also increased resistance to the BRD4 inhibitor JQ1, in line with upregulation of the BRD4 target MYC in these cells (Fig. 5m and Supplementary Fig. 6m). These results document that CHMP5 contributes to T-ALL chemoresistance mechanisms.

### CHMP5 is required for T-ALL initiation in vivo

Finally, we investigated whether CHMP5 is required for T-ALL initiation using a murine bone marrow (BM) chimera model wherein c-Kit+-enriched BM progenitors cells transduced with ICN1 initiate a lethal CD4+CD8+ T-ALL disease[63]. We specifically transplanted wildtype (WT) congenic (CD45.1+) recipient mice with ICN1-transduced (CD45.2+) BM donors from *Chmp5*[f/f]*Cd4*-Cre[-] ("WT") or *Chmp5*[f/f]*Cd4*-Cre[+] ("KO") mice in which Cre recombinase expression is driven by the *Cd4* promoter and *Chmp5* deletion is restricted to CD4-expressing thymocytes[27]. Transduction efficiency (>90%) of donors cells was confirmed by expression of a truncated nerve growth factor (NGFR) protein co-expressed by the bi-cistronic retroviral vectors encoding ICN1 and leukemia was monitored in the blood of animals by flow cytometry (Supplementary Fig. 7a–d). Of note, like human T-ALL, primary CD4+CD8+ (double positive, DP) T-ALL cells in this murine T-ALL model expressed much higher amounts of CHMP5 than normal DP

thymocytes (Supplementary Fig. 7e), suggesting that upregulation of CHMP5 in T-ALL cells is likely driven by oncogene activity.

BM chimera mice generated with ICN1-transduced WT donors as expected[63] developed a fully penetrant lethal (median survival, ~5 weeks) T-ALL characterized by marked leukocytosis comprising T-ALL (i.e., CD45.2+NGFR+) cells, splenomegaly with disrupted white and red pulp zones, and substantial leukemic cell infiltration of tissues (Fig. 6a–d and Supplementary Fig. 7f, g). In contrast, chimera mice generated with ICN1-transduced KO donors survived and did not show detectable T-ALL symptoms up to 12 weeks after transplantation (Fig. 6a–d and Supplementary Fig. 7f, g). Immunophenotyping of BM, spleen and peripheral blood confirmed a marked depletion of T-ALL (i.e., CD45.2+NGFR+) cells in KO chimeras, unlike WT chimeras that harbored large numbers of CD45.2+NGFR+ T-ALL cells (Fig. 6d and Supplementary Fig. 7f). Moreover, WT chimera T-ALL cells were phenotypically CD4+CD8+, whereas the few KO CD45.2+NGFR+ cells had reduced frequencies of CD4+CD8+ cells and also contained CD4-CD8+ and CD4-CD8- cells (Supplementary Fig. 7h). Together, these results demonstrate an essential requirement for CHMP5 in ICN1-initiated T-ALL development in vivo.

To understand the mechanisms by which CHMP5 promoted T-ALL initiation, we performed RNA-seq on sorted splenic CD45.2+NGFR+ (i.e., ICN1-transduced) cells from WT and KO T-ALL mice. Analysis of DEGs (Log2fold-change ≥ 1.2; adjusted $P < 0.05$) between WT and KO-derived CD45.2+NGFR+ cells revealed 1157 upregulated and 706 downregulated genes. Similar to CHMP5-depleted human T-ALL cells (Fig. 1), pathway analysis of these DEGs revealed downregulation of "MYC targets" in KO CD45.2+NGFR+ cells (Fig. 6e, f). Accordingly, KO CD45.2+NGFR+ cells expressed significantly less *Myc* and metabolic and cell cycle MYC target genes including *C1qbp, Ldha, Phb2* and *Cdk4* (Fig. 6g and Supplementary Fig. 7i).

To track changes in MYC protein expression in our murine T-ALL model, we introduced a MYC-GFP fusion protein knock-in reporter allele[64] into donor mice, allowing us to quantify MYC protein expression by flow cytometry of GFP fluorescence even in very few cells. Consistent with *Myc* mRNA downregulation in these cells, KO CD45.2+NGFR+ cells contained significantly fewer MYC (GFP+) cells that also expressed significantly lower levels of MYC measured by GFP mean fluorescence intensity (MFI) protein (Fig. 6h). These results are consistent with CHMP5 promoting the BRD4-p300-MYC axis that induces MYC expression in ICN1-driven T-ALL[13–15]. Interestingly, unlike ICN1-induced DP (T-ALL) thymocytes, normal DP thymocyte generation does not require CHMP5[27] (Supplementary Fig. 7j). This suggests that DP thymocyte dependency on CHMP5 is likely imposed by NOTCH1 (ICN1) to enable p300-BRD4-driven transcription of T-ALL genes. Accordingly, while normal DP thymocytes did not express the MYC-GFP reporter, ICN1+ DP (T-ALL) express MYC-GFP in a CHMP5-dependent manner (Supplementary Fig. 7k). Moreover, stimulation with recombinant delta-like 4 (DLL4)-Fc NOTCH1 ligand[65] induced MYC in WT but not CHMP5-deficient (*Chmp5*[f/f]*Cd4*-Cre[+]) DP thymocytes (Supplementary Fig. 7l).

We took advantage of the MYC-GFP reporter to enumerate the specific impact of CHMP5 deficiency on leukemia-initiating cells (LIC) wherein LICs in ICN1-driven T-ALL express CD34 and MYC[13,66]. CD45.2+NGFR+ cells from KO chimera animals were not only depleted of MYC-GFP[hi] CD34+ cells, but they also lacked any cells expressing CD34 in their BM and spleen (Fig. 6i). This suggest that impairment of T-ALL development from KO progenitors likely resulted from LIC abrogation. Collectively, these in vivo results unveil an essential requirement for CHMP5 in T-ALL initiation and reinforced the CHMP5-dependent regulation of the BRD4-p300-MYC axis which promotes the NOTCH1-driven T-ALL program. CHMP5's function and targets appear to be conserved in mice and humans as DEGs in ICN1-induced murine T-ALL and human NOTCH1-dependent T-ALL lacking CHMP5 include several overlapping genes and pathways highlighted by *MYC*

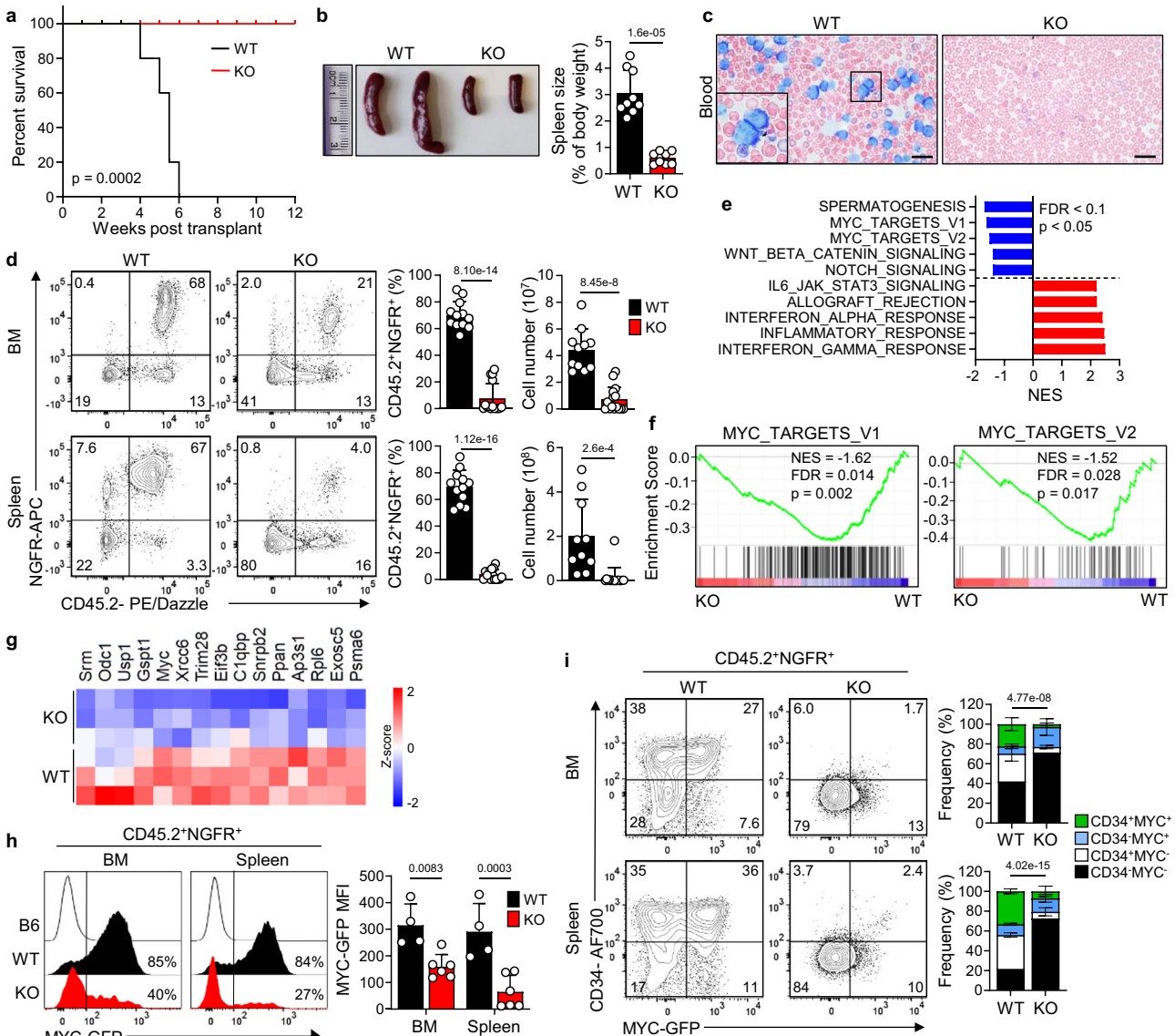

**Fig. 6 | CHMP5 deficiency impairs T-ALL development and progression in vivo.**
**a** Kaplan–Meier survival curve of leukemia mice generated by injection of ICN1-
NGFR-transduced *Chmp5^f/f^Cd4*-Cre⁻ (WT) or *Chmp5^f/f^Cd4*-Cre⁺ (KO) c-Kit-enriched
bone marrow (BM) cells. *p*, Mantel–Cox log-rank test; *n* = 9 mice/group. **b** Spleen
images and mean ± SD spleen weights. Student's *t* test, two-tailed. WT: *n* = 9 mice,
KO: *n* = 8 mice, from 2 independent experiments. **c** Wright-Giemsa staining of blood
from leukemia mice at 4 weeks post-transplant. Scale bar = 20 μm. **d** Representative
flow cytometry plot of CD45.2 and NGFR expression on BM (top) and spleen
(bottom) cells at 4 weeks after BM transplantation. Mean (±SD) of CD45.2⁺NGFR⁺
cell frequency (%) and absolute cell numbers are shown in graphs (right). Sample
size: frequency, WT, *n* = 12, KO *n* = 15; numbers, WT, *n* = 11, KO, *n* = 15 experimental
mice per group. Student's *t* test, two-tailed. **e** Hallmark pathway enrichment scores
of DEGs (fold-change ≥1.2; adjusted *p* < 0.05) in splenic WT vs KO CD45.2⁺NGFR⁺

cells at 4 weeks post-transplant (*n* = 3 mice per group). *p*-value determined by
Weighted Kolmogorov–Smirnov test and adjusted for multiple comparisons.
**f** GSEA plots of Hallmark MYC-target pathways from panel (**e**) (*n* = 3 mice per
group). p-value determined by Weighted Kolmogorov–Smirnov test and adjusted
for multiple comparisons. **g** Heatmap of DEGs from the MYC-target pathways. All
genes, have a *p* < 0.05. **h** MYC-GFP expression on CD45.2⁺NGFR⁺ cells from mice
with percentage of MYC-GFP⁺ cells indicated (left) and mean ± SD GFP mean
fluorescence intensity (MFI) (right). 2-way ANOVA. WT: *n* = 4; KO: *n* = 6 mice.
**i** Representative flow cytometry plot of CD34 versus MYC-GFP expression on
CD45.2⁺NGFR⁺ cells with mean ± SD frequencies of each gate in BM (top) and spleen
(bottom). 2-way ANOVA. WT: *n* = 4; KO: *n* = 6 mice. Source data are provided as a
Source Data file.

(Supplementary Fig. 7m, n) and genes that were upregulated in
NOTCH-dependent *MYC* super-enhancer (N-Me)-deficient T-ALL cells
(which have diminished MYC expression)[17] were also upregulated in
the KO T-ALL transcriptome (Supplementary Fig. 7o).

## Discussion

Using human and murine T-ALL models we have uncovered a critical
requirement for the ESCRT protein CHMP5 in promoting the T-ALL
initiation and maintenance gene program. This function of CHMP5 was
at least in part driven by its ability to potentiate p300-BRD4 interaction

to mediate H3K27ac of enhancers and super-enhancers of key T-ALL
genes (Supplementary Fig. 8). The significance of CHMP5 function in
T-ALL pathogenesis was supported by several lines of evidence: (i)
oncogenic ICN1, which in wildtype BM progenitors initiated a lethal
CD4⁺CD8⁺ T-ALL disease in mice, failed to cause T-ALL in BM pro-
genitors where CHMP5 is selectively deleted in CD4-expressing thy-
mocytes; (ii) CHMP5 deficiency in activated NOTCH1-driven human
T-ALL impaired the T-ALL gene program exemplified by *MYC* and
phenocopied the compromised metabolic fitness of MYC-deficient T-
ALL cells; (iii) compared to normal T cells, CHMP5 is highly expressed

across T-ALL subtypes and higher expression of CHMP5 correlated with worse T-ALL patient survival; (iv) CHMP5 deficiency synergistically improved chemotherapy efficacy, implicating CHMP5-driven mechanisms in chemoresistance in T-ALL. Together, our findings highlight a fundamental mechanism of T-ALL pathogenesis in which the ESCRT protein CHMP5 functions as a positive regulator of the BRD4-p300 dependent transcription of T-ALL genes, including MYC, a key transcription factor that enables T-ALL initiation and maintenance of NOTCH1-driven T-ALL.

High MYC and CD34 expression define T-ALL cells with leukemia-initiating cell (LIC) properties in activated NOTCH1-driven T-ALL[13,66]. LIC activity is abolished in *Myc*-deficient T-ALL cells and MYC-GFP[lo/-] murine T-ALL cells expressing the same MYC-GFP fusion reporter used in our study failed to cause disease in secondary recipients[13]. That CHMP5 deficiency caused severe depletion of all CD34[+] cells in primary murine T-ALL, including loss of the LIC-enriched CD34[+]MYC-GFP[hi] fraction, indicated that failure of CHMP5-deficient thymocyte progenitors to support ICN1-initiated T-ALL was likely due to impaired generation of LICs. This data supports a mechanism in which CHMP5 promoted T-ALL initiation at least in part by a BRD4-MYC-driven LIC transcriptional program. As LICs not only mediate but also sustain leukemia regeneration that promotes relapse[13], our findings also suggest that therapeutic depletion of CHMP5 has the potential to achieve durable T-ALL suppression.

Whether and how ESCRT proteins contribute to tumorigenesis has become of significant interest given their function in membrane remodeling is required for many key cellular processes. However, studies implicating ESCRT proteins in cancer have largely attributed their activity to membrane remodeling and repair that limits cell death[67–70]. Instead, and a fundamental insight from our study, we discovered a nuclear, chromatin-bound fraction of CHMP5 which, like other ESCRT proteins, was previously considered to be cytosolic[25]. In line with this nuclear localization, CHMP5 contained an N-terminal bipartite NLS in CHMP5 that appears to be highly conserved across jawed vertebrates in which it first appeared, suggesting that CHMP5's nuclear function was acquired later in evolution. How cytosolic versus nuclear CHMP5 trafficking is regulated is presently unclear but the presence of an NLS in CHMP5 suggest that it is actively imported into the nucleus. Importantly, our data support a model in which nuclear and chromatin-bound CHMP5 is the essential factor of the transcriptional machinery that promotes p300-BRD4-dependent transcription of T-ALL genes. Nevertheless, it remains possible that ESCRT-dependent functions of CHMP5 not evaluated in our study, potentially dependent on the putative NLS residues, also contribute to T-ALL pathogenesis.

Previous reports from a yeast-two-hybrid screen predicted ESCRT protein interaction with nuclear proteins[71] and some ESCRT proteins have been reported to promote nuclear envelope remodeling[72–74]. Only ESCRT protein CHMP1A has been demonstrated to interact with chromatin-binding proteins in the nucleus[75]. Importantly however, CHMP5 and CHMP1A appear to play distinct roles in the nucleus and in T-ALL pathogenesis. Whereas CHMP1A, which also has a bipartite NLS, functioned to recruit Polycomb-group proteins to silence genes[75], CHMP5 promoted transcription by facilitating BRD4-p300 recruitment to enhancers and super-enhancers. In addition, *CHMP5* transcript levels, but not *CHMP1A* or *VPS4A*, were prognostic in T-ALL patients.

As T-ALL cells expressed significantly more CHMP5 proteins than normal peripheral T-cells or DP thymocytes, our results further suggest that one mechanism by which oncogenes in T-ALL might hijack the transcriptional machinery to selectively control transcriptional output is by increasing CHMP5 expression. Higher amounts of CHMP5 would have the effect of increasing p300 interaction with BRD4, an outcome that would amplify their recruitment to enhancer and super-enhancer elements at key T-ALL genes. Because p300 catalyzes H3K27ac of *cis* regulatory enhancers[46–48] and BRD4 promotes p300's HAT activity[47], this p300-BRD4 interaction conceivably operates as a

positive feed-forward loop leading to further enriched BRD4 occupancy at enhancers and super-enhancers. In T-ALL cells, this process appears to be critically dependent on CHMP5.

P300 is pro-tumorigenic in blood cancers like acute myeloid leukemia (AML)[76–78] but its role in T-ALL is poorly understood. Our findings that impaired T-ALL initiation and maintenance in CHMP5-depleted T-ALL cells was at least in part due to disruption of the p300/BRD4 activity suggest a pro-tumorigenic role for p300 in T-ALL pathogenesis. Therefore, it would be of interest to evaluate both p300 and BRD4 inhibitors as therapeutic agents against T-ALL, especially T-ALL with activating NOTCH1 mutations that display high BRD4 dependency[13–15]. In "proof-of-concept" data, we show indeed that a dual p300/BRD4 inhibitor NEO2734[52,53] was superior to p300 or BRD4 inhibitors alone in killing T-ALL cells, paving way for future evaluation of NEO2734 in T-ALL.

Surprisingly, despite impaired p300 binding at T-ALL genes like *MYC*, total p300 proteins were upregulated in CHMP5-depleted T-ALL cells. While the molecular basis and consequence of p300 upregulation in these cells are unclear and will require future studies, we speculate that it is likely due to its diminished interaction with BRD4, since BRD4 depletion similarly increased p300 proteins in CHMP5-sufficient cells. When associated with BRD4, it is possible that BRD4's kinase (or HAT) activity 'primes' p300 for degradation, analogous to regulation of MYC protein stability by BRD4-mediated phosphorylation[79]. Because p300 is known to "redistribute" to other genomic sites especially after its inhibition[80,81], it is possible that CHMP5 depletion causes p300 proteins to redistribute to regulatory elements of an alternate set of genes like those upregulated in CHMP5-depleted T-ALL. In this regard, it is noteworthy that prostate cancer cells in which p300 is inhibited upregulated similar pathways, including interferon signaling[50], like CHMP5-depleted T-ALL cells.

In conclusion, this study has uncovered a fundamental mechanism of T-ALL pathogenesis whereby transformation of thymocytes by oncogenes like activated NOTCH1 (ICN1), as well as T-ALL maintenance depend on the ESCRT protein CHMP5. High CHMP5 expression has also been reported in hepatocellular carcinoma[82] and AML where its loss increased AML cell apoptosis[83,84]. Whether CHMP5 also regulates the BRD4-p300-MYC axis and super-enhancers in these cancers remains to be determined. Therefore, future studies to elucidate CHMP5 function have the potential to advance our understanding of transcriptional addiction mechanisms in cancer cells.

## Methods

### Ethics statement

This study complied with all relevant ethical regulations and experimental protocols were approved by the appropriate review boards as indicated. Protocols for animal experiments were approved by Institutional Animal Care and Use Committees (IACUC) in accordance with federal regulatory requirements and standards accredited by AAALAC International.

### Mice

Six to eight-week-old male or female B6.SJL-Ptprca Pepcb/BoyJ (B6.SJL), NOD.Cg-Prkdc[scid] Il2rg[tm1Wjl]/SzJ (NSG), B6;129-Myc[tm1Slek]/J (MYC-GFP)[64], and B6.Cg-Tg(Cd4-cre)1Cwi/BfluJ (*Cd4*-Cre)[85] mice were purchased from Jackson Laboratory. *Chmp5*[fl/fl] mice (with loxP-flanked exons 3-7 of *Chmp5*) have been previously described[32]. Mice were maintained in specific-pathogen-free facilities under controlled ambient conditions (temperature, 20–26 °C; relative humidity, 30–70%) on a 12-h light/dark schedule in individually vented cages with food and water provided ad libitum. Mice and animal facility conditions were monitored daily by trained technicians and veterinary staff. Where applicable, endpoint was achieved when mice showed any three of the following signs: lethargy, hunching, piloerection and/or >15% loss in body weight at start of experiments. When required mice were

euthanized by carbon dioxide and cervical dislocation as approved by institutional IACUC. All animal experiments were performed under IACUC-approved protocols at Case Western Reserve University (Protocol #2017-0055) and the National Cancer Institute-Frederick (Protocol #23-066).

## Murine cell isolation and culture

Bone marrow, thymus, and spleen were harvested and processed by mechanical dissociation to obtain single cells. Blood was collected either from the tail vein or medial saphenous vein into K2- EDTA coated collection tubes (BD Biosciences #365974). Splenocytes, blood, and bone marrow cells were treated with ACK lysis buffer (Gibco #A1049201) to lyse red blood cells. Single-cell suspensions were filtered through 40-μm strainers and resuspended in complete RPMI (RPMI 1640 supplemented with 2 mM L-glutamine, 1 mM sodium pyruvate, 10 mM HEPES, pH 7.4, 1 × MEM nonessential amino acids, 50 IU/ml penicillin, 50 μg/ml streptomycin, 55 μM β-mercaptoethanol, and 10% FBS). For NOTCH1 activation, double-positive thymocytes were cultured overnight on plates coated with 5 μg/ml of recombinant mouse DLL4-Fc (Novus Biologicals, #10089-D4-050).

## Primary human cells and culture

Normal human T-cells were isolated from peripheral blood mononuclear cells (PBMC) obtained from the Case Western Reserve University Hematopoietic Biorepository & Cellular Therapy Core. PBMCs were separated from the blood using a Ficoll density gradient (GE Healthcare #17144002). CD4 or CD8 cells were isolated using negative selection Miltenyi Biotec kits. Primary human T-ALL samples (Supplementary Table 1) were obtained from the University of Pennsylvania Stem Cell and Xenograft Core, RRID: SCR_010035. Samples were collected from patients after written informed consent under protocols approved by the University of Pennsylvania IRB (#703185).

## Patient-derived primary T-ALL

Primary patient samples were thawed in 30 ml IMDM with 2.5% fetal bovine serum (FBS) and 10 μg/ml DNase, filtered and washed in PBS. At least 1 million live cells were retro-orbitally injected into NSG mice. Mice were bled by the tail vein to monitor leukemia progression by human CD45 expression. Patient-derived T-ALL cells were isolated from the spleen and bone marrow of mice by negative selection with mouse CD45 microbeads (Miltenyi Biotec #130-052-301) for experiments.

## Cell lines and culture

HEK293T (ATCC) and Plat-E cells (Cell Biolabs #RV-101) were cultured in DMEM supplemented with 10% FBS, 2 mM L-glutamine, 1 mM sodium pyruvate, 10 mM HEPES, pH 7.4, 1 × MEM nonessential amino acids, 50 IU/ml penicillin, and 50 μg/ml streptomycin. CUTLL1 (gift from Adolfo Ferrando, Columbia University) and primary T-ALL samples were cultured in complete RPMI (RPMI 1640 supplemented with 2 mM L-glutamine, 1 mM sodium pyruvate, 10 mM HEPES, pH 7.4, 1 × MEM nonessential amino acids, 50 IU/ml penicillin, 50 μg/ml streptomycin, 55 μM β-mercaptoethanol) with 20% FBS. Jurkat, Loucy, MOLT-3, MOLT-4, CEM, and SUP-T1 were purchased from ATCC; KOPT-K1 and DND-41 cells were provided by W. Pear (University of Pennsylvania) and cultured in complete RPMI with 10% FBS. HSB-2 was purchased from Sigma Aldrich and cultured in complete IMDM with 10% FBS. All cell cultures were maintained at 37 °C in 5% $CO_2$ incubators. For TCR downregulation experiments, CUTLL1 T-ALL cells were stimulated for 30 min on plates coated with 5 μg/ml human anti-CD3 (Biolegend #317326) and 1 μg/ml human anti-CD28 (Biolegend #302914). All cell lines were obtained from commercial vendors where they are validated for authenticity and were also routinely authenticated by PCR and confirmed to be Mycoplasma free during our studies.

## Bone marrow transduction and transplantation

Bones were processed with ACK and a mortar and pestle. Cells were then washed with complete RPMI and filtered through a 40 μm strainer. Bone marrow (BM) from CD45.2 mice was subjected to Lineage depletion (Miltenyi Biotec #130-110-470). CD45.2 BM cells were plated at $1 × 10^6$ cells/ml in a 10 cm plate in 10 ml of complete StemPro-34 SFM (ThermoFisher #10639011) containing recombinant mouse cytokines (20 ng/ml IL-3, 50 ng/ml IL-6, 10 ng/ml Flt3L and 50 ng/ml SCF, PeproTech), 2 mM L-glutamine, 50 IU/ml penicillin, 50 μg/ml streptomycin and 55 μM β-mercaptoethanol. After 24 h, cells were infected with retroviral supernatants on Retronectin coated plates in the presence of 8 μg/ml polybrene and centrifugation at 1,000xg for 2 h at 25 °C. After 48 h, BM from CD45.1+ mice for hemogenic support were subjected to CD3 depletion (Miltenyi Biotec #130-094-973) and cells were harvested and washed with PBS. 1 million CD45.2 and 250,000 CD45.1 cells were injected retro-orbitally into lethally irradiated (1000 rads) CD45.1 mice. Mice were monitored for leukemia by flow cytometry of the blood and euthanized at endpoints following IACUC-approved protocols as described above.

## In vitro drug treatments and viability assay

For viability assays, T-ALL cells were cultured for 3 days, except otherwise indicated, with the indicated concentrations of drugs and analyzed with the Vybrant MTT Cell Proliferation Assay Kit (Thermofisher Scientific, V13154). The following chemotherapy drugs were used: Cytarabine (AraC; #S1648), Vincristine (#S9555), JQ1 (#S7110), A485 (#S8740) and CCS1477 (#S9667) were purchased from Selleck Chemicals; NEO2734 (MedChemExpress, #HY-136938), and Compound E (Enzo Life Sciences, #ALX-270-415-C250). Where applicable, cells were treated with 1 μM dexamethasone (Selleck Chemicals, #S1322) for 18 h. For BRD4 degradation, cells were treated with the PROTAC MZ1 (MedChemExpress; # HY-107425) at 100 nM in complete media for 4 h at 37 °C.

## Flow cytometry analysis and cell sorting

Single-cell suspensions were washed with flow-cytometry buffer (PBS, 2% FBS and 1 mM EDTA) and stained with fluorochrome-conjugated antibodies for 30 min at 4 °C in the dark. After washing, cells were resuspended in a flow-cytometry buffer and DAPI (4′,6-diamidino-2-phenylindole, 0.5 μg/ml; ThermoFisher #D1306) solution to exclude dead cells. MitoSOX (M36008), MitoTracker (M46753), ER Tracker (E34250), and Cell ROX (C10422) were acquired from ThermoFisher and used as indicated in the manufacturers protocol. Click-iT Plus OPP Alexa Fluor 647 Protein Synthesis Assay kit (Invitrogen #C10458) was used as per manufacturer's instructions. Annexin-V was stained using Annexin-V APC (Biolegend #640941) and Annexin-V Binding buffer (Biolegend #422201) incubated with 7-AAD (Tonbo Bioscience #13-6993-T500) for 20 min at RT. FITC-VAD-FMK (Invitrogen #88-7003) was used at a 1:2000 dilution of complete medium and stained for 15 min at 37 °C. Cell cycle analysis was performed using the Click-iT Plus EdU Alexa Fluor 647 Flow Cytometry Assay Kit (Invitrogen #C10634). Cells were incubated for 2 h with 10 μM EdU and stained with LIVE/DEAD™ Fixable Violet Dead Cell Stain Kit (Invitrogen #L34955) before fixation. After the Click-chemistry was performed, samples were incubated with 7-AAD (BD Biosciences #559925) for 10 min before acquisition. Cell sorting was performed on a BD FACS Aria or Aria-SORP and cell analysis on a BD LSR Fortessa and BD LSR-II. Flow analysis was done using FlowJo software. The antibodies used are listed in Supplementary Table 2.

## Plasmids

C-terminal Flag-tagged CHMP5 (NP_084090.1) or N-terminal HA-tagged CHMP5 was amplified by PCR and cloned into pcDNA3.1 (ThermoFisher #V79020) or pHAGE (Harvard) vectors. BRD4 overexpression plasmid for HEK293T transfection was purchased from

Addgene (p6344 pcDNA4-TO-HA-Brd4FL #31351). shRNA constructs in a pLKO.1 vector targeting human CHMP5 were purchased from Sigma Aldrich along with 2 control non-targeting shRNAs. shRNAs targeting human MYC were a gift from Xi Chen (Baylor College of Medicine). shRNA sequences can be found in Supplementary Table 3. ICN1-NGFR (pMIG) in which ICN1-transduced cells are marked by NGFR expression, was gifted to us by Lan Zhou (CWRU). The human CHMP5 ΔNLS and point mutants were synthesized by Azenta Life Sciences.

## Virus generation and cell transduction

Lentiviruses were generated by transfecting HEK293T cells with the plasmid of interest along with VSVg and delta 8.9 packaging plasmid vectors using Lipofectamine 3000 Transfection Reagent (Thermo #L3000075). After 48 h, viral supernatants were harvested, filtered through a 0.45 μm filter, concentrated with an Amicon Ultra-15 centrifugal filter (EMD Millipore #UFC903024) and stored at -80 °C for future use.

Retroviruses were produced by transfecting Plat-E cells with the plasmid alone using Lipofectamine 3000. After 48 h, viral supernatants were harvested and filtered, then concentrated using Retro-X concentrator overnight (Takara Bio #631455). Untreated tissue culture plates were coated with 25 μg/ml RetroNectin (Takara Bio #T100B) overnight at 4 °C to be used for retroviral transduction. Plates were blocked the next day with 2% bovine serum albumin in PBS for 30 min at room temperature. Concentrated retrovirus was spun down on the RetroNectin-coated plates for 2 h before adding the cells. Cells were transduced with lentiviruses or retroviruses by adding 8 μg/ml polybrene (Sigma Aldrich #TR-1003-G) and a previously determined amount of virus based on titers in a 6 well plate. The cells were then spun at 1000xg for 2 h at room temperature and cultured for 48 h.

For the generation of shRNA knockdown cells, 48 h after transduction with shRNA lentivirus, cells were selected with 1 μg/ml puromycin (Invivogen #ant-pr-1) for 3-5 days. Deletion was validated by western blot and qPCR.

## RNA isolation and quantitative real-time PCR

Total RNA was isolated from all cells using TRIzol reagent (Thermofisher, #15596026) and 100 to 500 ng of RNA was reverse transcribed using the high-capacity cDNA reverse transcription kit (ThermoFisher, #4368814). Real-time qPCR were performed using Fast SYBR Green Master Mix (Applied Biosystems #43-856-12) and ran on QuantStudio 6Flex real-time PCR system (Applied Biosystems # 4485691). Individual gene expression levels were calculated using the change in cycling threshold ($\Delta C_T$) method as $2^{-\Delta C_T}$, where $\Delta C_T$ is $[C_T$ (gene of interest)- $C_T$ (housekeeping gene)]. qPCR primer sequences are included in Supplementary Table 4.

## Immunoprecipitation and immunoblots

Cells are harvested and washed with ice-cold PBS by centrifuging at 2,400xg for 5 min at 4 °C. Pellets were either stored at -80 °C or lysed immediately; for western blots, cells were lysed with RIPA (25 mM Tris-HCl pH 7.6, 150 mM NaCl, 1% NP-40, 1% sodium deoxycholate, 0.1% SDS) buffer. For co-immunoprecipitations, cells were lysed with IP buffer (20 mM Tris-HCl, pH 8.0, 150 mM NaCl, 5 mM MgCl₂, 0.5% NP-40). All lysis buffers were supplemented with 1X phosSTOP (Sigma Aldrich #4906837001) and 1X EDTA-free Protease Inhibitor Cocktail (Sigma Aldrich #4693159001). Lysates were collected after centrifuging cells at 16,200xg for 12 min at 4 °C. Cell pellets used for whole cell fractionation were fractionated using the Nuclear Complex Co-IP kit from Active motif (#54001) according to their protocol. Nuclear fractionation in soluble and chromatin-bound fractions was done by first lysing the cytoplasm with 0.1% NP-40 and 2 pop-spins for 10 s each. The nuclear pellet was then lysed with 3 times the volume of buffer A (10 mM HEPES, 1.5 mM MgCl₂, 10 mM KCl, 1 mM DTT, 0.5 mM PMSF, 1x protease inhibitor cocktail, 0.5% NP-40, 75 mM NaCl) for 10 min. After

centrifuging at 5000 g for 5 min, supernatants were taken as the non-chromatin bound (soluble) fraction while pellets were lysed with 4-times the volume of high salt buffer (10 mM HEPES, 20% glycerol, 350 mM NaCl, 1.5 mM MgCl₂, 0.4 mM EDTA, 0.5% NP40, 1 mM DTT, 0.5 mM PMSF, 1x protease inhibitor cocktail) for 30 min at 4 °C. After centrifugation at 12,000 g for 10 min, the supernatant is collected for the chromatin-bound fraction. Equal amounts of protein from each fraction was loaded for western blots.

Protein concentration was determined using Pierce™ BCA® Protein Assay Kits (ThermoFisher, #23225). 10–40 μg of protein was used for western blots, 200–500 μg of protein was used for immunoprecipitation. Immunoprecipitations were performed using magnetic Protein G Dynabeads (ThermoFisher #10004D) with incubation of the primary antibody done overnight at 4 °C with overnight rotation. NuPAGE LDS sample buffer (ThermoFisher #NP0007) and Sample Reducing Agent (ThermoFisher #NP0009) were added to all lysates and boiled at 70 °C for 10 min. Samples were then run on NuPAGE 4–12% Bis-Tris Protein Gels (ThermoFisher #NP0335BOX) and transferred to a PVDF membrane (EMD Millipore #IPVH00005). Blots were probed with primary antibodies overnight at 4 °C in 5% blocking solution. After washing and 1 h incubation with HRP-linked secondary antibodies, blots were developed using Pico- or Femto- chemiluminescent substrate (ThermoFisher) and visualized with autoradiography or digital imaging. Where indicated, band intensities were quantified by densitometry using the ImageJ program (NIH).

Cell-free protein interaction assays were performed as previously described[8]. Recombinant CHMP5 was purchased from Abcam (#ab134604), recombinant BRD4 was previously generated[8,86], recombinant p300 was purchased from Active Motif (#81158). Recombinant proteins were diluted with Buffer D (20 mM HEPES, 100 mM KCl, 0.2 mM EDTA, and 20% vol/vol glycerol) to 0.1 mg/ml and 0.5 μg of each were incubated either individually or mixed in a 1:1 ratio in 500 ml of TBS (50 mM Tris-HCl, 150 mM NaCl) with protease inhibitors. Proteins were incubated overnight at 4 °C with gentle rotation. Subsequently, 1 μg of anti-FLAG (Sigma #F1804) or anti-BRD4 (Bethyl Laboratories #A301-985A50) was incubated with the proteins for 4 h at 4 °C with gentle rotation, and 40 ml of magnetic Protein G Dynabeads (Thermo #10004D) was added to each condition and incubated for 2 h at 4 °C with rotation. The beads were then washed 3 times with TBS + 0.2% NP-40 and boiled at 70 °C for 10 min in TBS with NuPAGE LDS sample buffer and reducing agent. The samples were run on NuPAGE 4–12% Bis-Tris Protein Gels. All antibodies used in these studies are listed in Supplementary Table 2. Note that in some samples the anti-CHMP5 polyclonal antibody (ThermoFisher Scientific, #PA5-63303) can detect the two h an CHMP5 isoform: the major 219 amino acid (UniProt: Q9NZZ3-1) and a smaller 171 amino acid (UniProt: Q9NZZ3-2) isoform.

## Confocal microscopy

T-ALL (CUTLL1 and PDX T-ALL) cells were washed with PBS and adhered on poly-prep slides (Sigma #P0425) at room temperature for 30 min. Cells were then fixed with 4% PFA for 30 min at room temperature. After washing with PBS, the cells were permeabilized for 10 min with PBS containing 0.2% Triton X-100 and blocked for 30 min with blocking solution (PBS containing 1% BSA, 2% goat serum, 0.1% triton x-100). The slides were washed and incubated with primary antibody (1-2 μg/ml anti-CHMP5 or anti-IgG) over night at 4 °C in the dark. After 3 washes, the slides were incubated with 2 μg/ml AF647 conjugated secondary antibody (Invitrogen #A32733) for 1 h at room temperature. After washing, slides were mounted with hard-set mounting solution with DAPI (VectaShield #H-1500-10).

## Quantification of CHMP5 by Immunofluorescence

Three-dimensional images (17 slices, 0.8 micron spacing) were acquired with Nikon Eclipse Ti2 microscope equipped with Yokogawa

CSU-W1 spinning disk confocal system and Visitron VS-Homogenizer beam shaping unit, using Nikon PlanApol 60x/1.40 oil objective and Hamamatsu Orca-Flash4.0 camera. Original 16-bit images were analyzed with ImageJ and "Intensity measurement in 3D segmented cells" custom macro (https://github.com/janwisn/Intensity-measurement-in-3D-segmented-cells). Individual cells were identified, and label signal was measured (after background subtraction) in 3D within nucleus and cytoplasm of each cell.

## Protein stability assays
Cells were cultured in the presence of 20 μM MG132 (Sigma #M7449) for 6 h unless indicated otherwise. For half-life experiments, cycloheximide (Sigma #01810) was used at 50 μg/ml. Cells were subsequently processed and subjected to immunoblotting as above. Protein expression was quantified by densitometry and half-life was calculated by normalizing protein levels during cycloheximide treatment to DMSO treated cells without cycloheximide.

## Seahorse metabolic flux assays
Extracellular acidification rate (ECAR) was performed under glycolysis stress test and oxygen consumption rate (OCR) was performed under mitochondrial stress test. Briefly, $1 \times 10^5$ CUTLL1 were plated per well on poly-D-lysine coated Seahorse XFe96 microplates (Agilent #101085-004) in XF RPMI medium (Agilent #103576-100) supplemented with 1 mM Sodium Pyruvate, 2 mM L-glutamine, and 25 mM glucose for the Mito Stress Test (Agilent #103015-100), or 2 mM L-glutamine for the Glycolysis Stress Test (Agilent #103020-100). After an hour incubation at 37 °C, OCR and ECAR were measured using a 96 well XF96 Extracellular Flux Analyzer (Agilent Technologies). Measurements were taken under basal conditions and following sequential addition of the drugs provided with the stress test kits. Fluoro-carbonyl cyanide phenylhydrazone (FCCP) was used at 1 μM final concentration. Basal ECAR was calculated from the samples in the Mito Stress Test medium containing glucose, and basal OCR was calculated from the samples in the Glycolysis Stress Test medium at the start of the assay.

## Histology
Organs were fixed in 4% paraformaldehyde overnight and transferred to 70% ethanol before embedding. Embedding, sectioning and H&E staining was done at the Cleveland Digestive Diseases Research Core Center. Blood smears were made using Wright-Giemsa stain (Sigma #WG16). Images were taken using an Olympus IX73 microscope.

## CUTLL1 RNA-seq sample processing, library preparation, and sequencing
RNA was isolated from CUTLL1 cells independently transduced with either shControl or shCHMP5 in triplicate using the Qiagen RNeasy kit. RNA was submitted BGI (Hong Kong, China) for sample QC, library preparation, and sequencing. RNA was quantified and integrity was checked using Agilent 2100 Bioanalyzer. Library preparation was done using the NuGEN Trio RNA-seq library preparation kit (Redwood City, CA). Libraries were sequenced with Illumina HiSeq X Ten PE150bp sequencing and 30 million reads per sample.

## Murine RNA-seq sample processing, library preparation, and sequencing
CD45.2⁺NGFR(ICN1)⁺ cells were sorted from the spleens of chimera mice at median survival time-point (4 weeks post-transplantation). RNA was isolated from these cells using the TRIzol reagent and submitted to Azenta Life Sciences LLC (South Plainfield, NJ, USA) for library preparation and Ultra-low RNA sequencing. Total RNA samples were quantified using Qubit 2.0 Fluorometer (Life Technologies) and RNA integrity was checked using Agilent TapeStation 4200 (Agilent Technologies). The SMARTSeq HT Ultra Low Input Kit was used for full-length cDNA synthesis and amplification (Clontech), and Illumina Nextera XT library was used for sequencing library preparation. Briefly, cDNA was fragmented, and adaptor was added using Transposase, followed by limited-cycle PCR to enrich and add index to the cDNA fragments. Sequencing libraries were validated using the Agilent TapeStation and quantified by using Qubit Fluorometer as well as by quantitative PCR (KAPA Biosystems). The sequencing libraries were multiplexed and clustered on a flowcell. After clustering, the flowcell was loaded on the Illumina NovaSeq 6000 instrument according to manufacturer's instructions. Sequencing was performed at 2×150 Paired End.

## RNA-seq data processing and gene set enrichment analysis
Sequence reads for human samples control (CT) and knock-down (KD) were aligned to GRCh38 reference sequence annotated with Gencode release 39 with Star aligner (v2.7.9a)[87] with --sjdbOverhang 100 and default parameters. Mouse wildtype (WT) and knockout (KO) samples were aligned to GRCm39 with Gencode release M27 with Star aligner (v2.7.9a) with --sjdbOverhang 150 and default parameters. STAR aligned reads gene counts were calculated with htseq (v0.11.4)[88] with default parameters and differential expression analysis performed with DEseq2 (v1.38.3)[89] software. Read counts were filtered to retain genes with at least 10 counts in the smallest group. Gene set enrichment analysis (GSEA) for the RNA-seq data was performed with GSEA 4.3.2[90] and MsigDB Hallmark gene set collection[91] and curated gene lists from Kim et al.[31] and Herranz et al.[17]. For the comparison of CHMP5-regulated genes in human and mouse T-ALL, genes were extracted for mapping using BioMart R[92] and Ensembl release 105[93]. Some DEGs are lost during the overlapping of human and mouse gene IDs.

## Chromatin Immunoprecipitation (ChIP)
Fixation, isolation of the chromatin, immunoprecipitation and DNA isolation for ChIP analysis was done using the SimpleChIP® Enzymatic Chromatin IP Kit (CST #9003) and ran according to its protocol. Sonication was performed for 15 s on, 45 s off for 15 min. 10 μg of chromatin was isolated with 4 μg of indicated antibodies overnight. Antibodies used in these studies are listed in Supplementary Table 2. qPCR was performed with 1ul of each per primer and compared to 2% input sample. Library preparation and sequencing of ChIP samples was performed by Azenta Life Sciences on the Illumina with 2 × 150 bp, ~350 M PE reads.

## ChIP-seq library preparation and sequencing
ChIP DNA samples were quantified using Qubit 2.0 Fluorometer (Life Technologies) and the DNA integrity was checked with 4200 TapeStation (Agilent Technologies). ChIP-Seq library preparation and sequencing reactions were conducted at Azenta US, Inc. (South Plainfield, NJ, USA). NEB NextUltra DNA Library Preparation kit was used following the manufacturer's recommendations (Illumina). Briefly, the ChIP DNA was end repaired and adapters were ligated after adenylation of the 3'ends. Adapter-ligated DNA was size selected, followed by clean up, and limited cycle PCR enrichment. The ChIP library was validated using Agilent TapeStation and quantified using Qubit 2.0 Fluorometer as well as real time PCR (Applied Biosystems). The sequencing libraries were multiplexed and clustered on one lane of a flowcell. After clustering, the flowcell was loaded on the Illumina NovaSeq 6000 instrument according to the manufacturer's instructions (Illumina). Sequencing was performed using a 2 × 150 Paired End (PE) configuration. Image analysis and base calling were conducted by the Control Software (NCS). Raw sequence data (.bcl files) generated from the Illumina instrument were converted into fastq files and de-multiplexed using Illumina's bcl2fastq 2.17 software. One mismatch was allowed for index sequence identification.

## ChIP-seq data processing

Sequencing reads from BRD4, RNA Pol II, and H3K27 acetylation ChIP from control (CT) and knockdown (KD) and the input read samples were aligned to the reference genome UCSC hg38 with Bowtie2 (v2.4.5)[88] with −local setting and unique reads were kept by filtering out unmapped, duplicated and multimapped reads with sambamba 0.8.2[94] custom filters -F "[XS] == null and not unmapped and not duplicate". MACS2 (v2.7.1)[95] with default parameters was used for peak calling over the corresponding background. MACS2 peaks were annotated with ChIPseeker (v1.34.1) with TSS defined from -1kb to +1kb[96]. Heatmaps and metaplots displaying ChIP-seq occupancy were created with ngsplot[97]. ChIP tracks were generated using UCSC genome browser. The ROSE software[44,98] was used to map typical and super enhancers with a maximum linking distance of 1 Mb for stitching.

## Traveling Ratio (TR) calculation

Traveling Ratio (Pausing Index) was calculated based on RNA Pol II ChIP-seq utilizing the PIC software (https://github.com/MiMiroot/PIC) as previously described in[99] with default settings, −longest option to get the longest isoform, and gene body defined as +300 bp to the end of annotated gene. Genes <1 kb were removed from the analysis. Traveling Ratio (TR) was calculated for all genes, activated genes (promoters with H3K27ac peaks), non-activated genes (no H3K27ac peak), BRD4 target (genes bound by BRD4 at promoters from ChIP-seq), and MYC target genes (HALLMARK MYC_TARGETS _V1 and V2). Traveling Ratio comparison between CT and KD samples was performed on log2 transformed ratios in R[100] using Welch two sample t-Test.

## Patient survival curves

RNA-seq and clinical data from TARGET-ALL-P2 cohort 1 was downloaded from the GDC data portal. Patients were divided into two groups according to the expression level of CHMP5, VPS4A, or CHMP1A; the top 20% were classified as highly expressed group, and the bottom 20% were classified as lowly-expressed group. The entire group is comprised of the lowly-expressed group and highly expressed group. Survival analysis is performed on these groups and survival curves are generated by using R-package 'Survival' and 'Survminer'.

## Statistics and reproducibility

Statistical analysis was performed using GraphPad Prism version 9.0 and later versions. Comparisons between groups were determined with a two-tailed, unpaired Student's $t$ test or a one/two-way ANOVA or as specified in figure legends. Differences were considered significant if $p < 0.05$, specific p-values are indicated on the graph or in figure legends. Statistics were not derived from technical replicates or with $n < 3$. All experiments were replicated at least two times, the exact number of times is indicated in the figure legends. Data that could not be reproduced was not included in the paper. No statistical tests were used to predetermine sample size and no data were excluded from the analyses. Sample sizes for mice experiments were based on previous experiments and experiments were performed with at least 5 mice per group and repeated 3 times with similar results. Recipient mice were housed together, irradiated, and randomly split into two groups for WT and KO bone marrow chimeras. Although blinding could not be used entirely, investigators analyzed mice by ear tag number, not group, to be unbiased in their assessments. Investigators were not blinded to allocation during experiments and outcome assessment for in vitro studies.

## Materials availability

Further information or requests for materials and resources reported in this manuscript should be directed to and will be fulfilled by the lead contact, Stanley Adoro (stanley.adoro@nih.gov) with a completed Materials Transfer Agreement when applicable.

## Reporting summary

Further information on research design is available in the Nature Portfolio Reporting Summary linked to this article.

## Data availability

All sequencing data have been deposited to the Gene Expression Omnibus database (GEO) and are publicly available under the SuperSeries GSE244200. RNA-seq on control (CT) and shRNA-mediated CHMP5 depletion (KD) CUTLL1 human T-ALL cells can be downloaded under GEO accession number GSE244198. ChIP-seq of BRD4, Pol II, and H3K27ac in CT and KD CUTLL1 cells can be downloaded under GEO number GSE244197. Murine RNA-seq on wildtype and *Chmp5*-deficient ICN1-transduced CD45.2[+]NGFR[+] splenocytes can be downloaded under GEO number GSE244199. The primary human T cells and human T-ALL[54,55] datasets used in this study are available in GEO database under accession numbers GSE33470 and GSE33469. The TARGET T-ALL data is from the Therapeutically Applicable Research to Generate Effective Treatments (TARGET) (https://www.cancer.gov/ccg/research/genome-sequencing/target) initiative, and is publicly available in the NCBI database of Genotypes and Phenotypes under accession code phs000464.v7.p1[56]. The BRD4 ChIP-seq used in Fig. 2o is publicly available in the NCBI Gene Expression Omnibus database under accession code GSE5180041[41]. Gene lists used for GSEA plots in Supplementary Fig. 1h and 6o are available in PMID: 25194570[17]. Gene lists used for GSEA plots reported in Supplementary Fig. 1i are available in PMID: 16116477[31]. Pediatric T-ALL patient publicly available data used in Supplementary fig. 6f are available in the National Omics Data Encyclopedia (NODE) under accession code OEP00000760[57]. The remaining data are available within the Article, Supplementary Information or Source Data file. Source data are provided with this paper.

## Code availability

Software codes for quantifying immunofluorescence images are available at https://github.com/janwisn/Intensity-measurement-in-3D-segmented-cells.

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

## Acknowledgements

We thank Drs. Anastasia Tikhonova for critical reading of the manuscript and helpful suggestions; Martin Carroll for providing primary human T-ALL samples; Adolfo Ferrando for CUTLL1 cells; Xi Chen for MYC expression plasmids; Warren Pear for KOPT-K1 and DND-41 cell lines. This work was supported by grants from the US Department of Defense CA180768-W81XWH1910306 (S.A.); National Cancer Institute (NCI) K22 CA218467 (S.A.); National Institute of Allergy and Infectious Disease R01 AI143992 (S.A.); American Cancer Society RSG-19-025-01-DDC (S.A.) and the Intramural Research Program of the National Institutes of Health, National Cancer Institute, Center for Cancer Research ZIA BC 012135 (S.A.). L.Z. is supported by grants from the National Heart, Lung, and Blood Institute R01 HL103827 and NCI R01 CA222064. K.U-W. was supported in part by a Cell and Molecular Biology Training grant from the National Institute of General Medicine T32 GM008056 to the Case Western Reserve University School of Medicine. We acknowledge use of the high-performance computational capabilities of the Biowulf Linux cluster at the NIH. Schematic diagrams were created with BioRender.com.

## Author contributions

Conceptualization, K.U.-W., and S.A.; methodology, K.U.-W., and S.A.; software, S.R., R.W., Q.T., Q.C., J.W., and D.M.; formal analysis, K.U.-W., B.N.D., D.S.S., and S.A.; investigation, K.U.-W., B.N.D., S.G.C., J.T., and S.A.; resources, L.Z. and D.S.S.; visualization, K.U.-W., S.R., and S.A.; writing—original and final draft, K.U.-W. and S.A.; writing—review & editing, K.U.-W. and S.A., with feedback from all authors; supervision, S.A.; funding acquisition, S.A.

## Funding

## Competing interests

The authors declare no competing interests.
