## [Transparent Peer Review file · Nature Communications]

The ESCRT protein CHMP5 promotes T cell leukemia by enabling BRD4-p300-dependent transcription

Corresponding Author: Dr Stanley Adoro

Version 0:

Reviewer comments:

Reviewer #1

(Remarks to the Author)

In this interesting article, Umpired-Wilson et al identify CHMP5 as a critical factor in T-ALL. Authors start by showing CHMP5 overexpressed in T-ALL compared to normal cells, and they show the CHMP5 knockdown in T-ALL cell lines sensitizes them to treatments with AraC or Dexamethasone+GSI. Gene expression profiling upon CHMP5 knockdown revealed a clear downregulation of the MYC transcriptional program. Mechanistically, authors show that CHMP5 interacts with BRD4 specifically in the nucleus and show multiple ChIPseq data supporting this interaction mediates PolII pause release and contributes to the formation of the BDME superenhancer. Further, CHMP5 promotes the interaction between BRD4 and p300, leading to increased H3K27 acetylation. Consistently, NOTCH1-induced T-ALL generation was completely abrogated in the absence of CHMP5, and CHMP5-null cells infected with NOTCH1 showed decreased levels of MYC and decreased LIC frequency. I would like to congratulate authors for their very nice study. Still, I have some comments that might further improve the paper (not listed in order of importance):

1. Authors go on to directly show the effects of CHMP5 knockdown in response to some chemotherapy (the rationale for the use of AraC or Dexa+GSI is somewhat unclear), but they do not show the effects of CHMP5 knockdown on its own, which is relevant, as this would be reflective of effects on T-ALL progression (as opposed to initiation shown in Fig 6). Authors should show proliferation, as well as cell cycle/apoptosis effects of shCHMP5 cells vs shControl cells. Ideally, authors could generate a tamoxifen-inducible CHMP5 cKO T-ALL (by breeding the floxed mice with tmx-Cre mice) in order to show the effects of secondarily deleting CHMP5 in already established T-ALL in vivo, but that might be beyond the scope of the present study.

2. It is unclear how CHMP5 knockdown increases NR3C1 expression. Could authors show any data suggesting a mechanism? if not, this should at least be discussed.

3. Authors state "CHMP5-regulated mechanisms dictate patient response to chemotherapy" (Line 144). This is a quite strong statement, I would suggest to down tone the "dictate".

4. In Fig S1B, authors show surprisingly low ICN1 levels for several cell lines, such as DND41, KOPTK1 or JURKAT. Specifically, JURKAT shows the same pattern as LOUCY in the next lane, and Loucy is a NOTCH1-negative cell line. Jurkat should definitely show more ICN1 than Loucy at the very least. I would suggest to repeat this WB and/or verify that all of their cell lines are indeed what they are supposed to be.

5. In Fig 2, authors show correlation of MYC mRNA/protein and CHMP5 protein in their own data. Can they also show this with CHMP5 mRNA? Related to this, it would be more relevant if authors could see this MYC/CHMP5 mRNA correlation in the T-ALL patient dataset from TARGET (Liu et al, Nat Genet, 2017).

6. Fig 2B and 6F show GSEA of MYC signatures. Even if significant, these plots are not as strong as they could be, which might be due to the fact that these are the MYC TARGET datasets from public databases. I would suggest authors to run their gene expression signature against the specific signature obtained upon deletion of the N-Me enhancer in T-ALL (Herranz et al, Nat Med, 2014), as this dataset is T-ALL specific and might give them a stronger correlation with the T-cell specific MYC-driven program.

7. In Fig 3C, cells lacking the NLS seem to be absent from the nucleus. Still, that FLAG WB image is way less exposed than the FL, making this comparison difficult. I would suggest to revise this WB so that the signal in FL and deltaNLS is similar, in order to better assess whether delta NLS is completely absent from the nucleus, or still translocates but to a lesser extent.

8. In Fig 4, authors show data supporting pol II pause release. However, they do this by looking at RNA pol II ChIPseq alone. This is better assessed by looking at the ration of pol II phospho-S5(TSS) vs pol II phospho-S2(TES). While performing ChIPseq for these too might be an overkill at this point, authors could perform local ChIP for this specific polII versions at MYC and/or other targets.

9. In Fig 4 it is slightly confusing the nomenclature used by authors when they just show/say "SE". Is this the BDME enhancer? I would suggest to consistently show in the figures both N-Me and BDME and avoid confusions. Related to this, I think it is relevant to show N-Me profile (BRD4, H3K27ac...) everywhere. N-Me is the primary regulator of the NOTCH1-MYC axis in T-cells, so it is important to show potential effects in all of their data, even if in Fig 3 they didn't show BRD4/CHMP5 binding there. I would be surprised if authors observed such drastic effects on MYC without N-Me being more affected (maybe H3K27ac data in Fig 4 and S4?). Related to this, N-Me loss has a strong effect on normal T-cell development (and MYC expression throughout it) but, according to authors, CHMP5 effects are milder. Authors state that CHMP5-null DP cells lack MYC (line 479) but i couldn't find this data here or on their Nat Immunol paper. Can authors show this? Maybe taking advantage of their MYC-GFP tag?

10. Authors fail to generate NOTCH1-induced leukemias, and authors conclude this is likely driven by lack of MYC expression. Can authors generate leukemias by overexpressing MYC in CHMP5-null progenitor cells?

Reviewer #2

(Remarks to the Author)

In this study, Umphred-Wilson et al. demonstrate an intriguing novel nuclear function for the protein CHMP5. CHMP5 has been characterized as a cytoplasmic factor involved in endosomal sorting and membrane remodeling (ESCRT machinery). Yet here the authors demonstrate with a wealth of data that CHMP5 has a nuclear localization sequence, can transport to the nucleus where it can associate with chromatin together with BRD4 and p300 to promote gene transcription. They also demonstrate in cell lines and primary patient samples that CHMP5 is upregulated in T-cell acute lymphoblastic leukemia compared to normal T cells, and so interaction with BRD4-p300 at enhancer elements contributes to MYC expression. Genetic knockdown of CHMP5 decreases the MYC expression program in T-ALL and can decrease T-ALL cell proliferation both in vitro and in vivo.

The preponderance of data included in the paper convincingly demonstrates CHMP5 involvement in transcriptional outputs in T-ALL. I have several minor questions that should be addressed:

- 1) The K-M data in Figure 1D and Ext. Data Fig 1D seem to be replicated, although it purports to show different cohorts defined by expression of either CHMP5 or a control gene.
- 2) It is interesting that CHMP5 KD can have a significant anti-MYC effect, with dramatically reduced cell proliferation (Fig 2J), and yet there is no observable apoptosis and very minimal cell cycle arrest phenotype. Are other antiproliferative mechanisms a consequence of CHMP5 loss?
- 3) Does overexpression of CHMP5 in normal T cells or CHMP5-low T-ALL induce BRD4 activity and MYC expression?
- 4) The authors claim that BRD4 signal is specifically lost at distal enhancers upon CHMP5 loss, and demonstrate this at selected loci - can they show this cumulatively across the epigenome?
- 5) There is a striking upregulation of p300 expression upon CHMP5 loss. Can the authors speculate as to the consequences of this? p300 is known to 'redistribute' to other regulatory sites in the epigenome upon inhibition - where does all the newly expressed p300 go?
- 6) The identity of the loading controls used in Ext Data Figure 3A is missing.

Reviewer #3

(Remarks to the Author)

The manuscript by Katharine Umphred-Wilson and colleagues presents compelling evidence implicating Chmp5, an ESCRT-III subunit, in the promotion of T-ALL. Through a comprehensive series of experiments conducted in both cellular and murine models, the authors propose a mechanism wherein a nuclear fraction of Chmp5 interacts with BRD4 and p300, to augment their recruitment to super-enhancers and subsequently upregulating the transcription of various T-ALL-associated genes, including MYC. The loss of Chmp5 leads to a significant decrease in MYC expression and other T-ALL genes, rendering Chmp5-depleted cells more susceptible to chemotherapy. While the findings are intriguing and the experiments well-executed, there are two critical points that require clarification to strengthen the conclusion that exclusively nuclear Chmp5, in conjunction with BRD4 and p300, governs the transcriptional effects on T-ALL genes and hence T-ALL pathogenicity.

Major Points:

(1) While the majority of experiments demonstrate the necessity of Chmp5 for T-ALL gene expression, it remains unclear whether solely nuclear Chmp5, in concert with BRD4 and p300, accounts for the observed effects. To better discern nuclear from cytosolic Chmp5 functions, such as the down-regulation of Notch signaling by ESCRTs or the stabilization of the pro-survival protein Bcl-2, we recommend generating point mutations in the nuclear localization signal (NLS) of Chmp5, preserving cytosolic function while impeding nuclear import. Alternatively, identifying mutations in Chmp5 that specifically disrupt its binding to BRD4 or p300 would be helpful. Expressing these mutant Chmp5 variants in Chmp5-deficient cells could elucidate the distinct roles of nuclear and cytosolic Chmp5. If these experiments exceed the study's current scope, the authors should tone down their conclusions and entertain the possibility of non-nuclear Chmp5 involvement in T-ALL gene expression.

(2) In prior research, the authors convincingly demonstrated Chmp5's necessity for T cell development, partially through Bcl-2 stabilization, with Chmp5 loss resulting in impaired T-cell development. Given these findings, it is perhaps unsurprising that mice lacking Chmp5 in T cells fail to develop T-ALL. Please explain how Chmp5's posttranslational role in T-cell development relates to its nuclear function in T-ALL development.

Minor Points:

(a) Please denote "Mr" for Western blotting.

(b) Chmp5 localization should be assessed via immunofluorescence microscopy.

(c) Western blot quantifications (e.g., Figure 3A) do not seem to align with the blots; it seems unlikely that 20% of Chmp5 in SUPT1 cells are nuclear.

(d) Please Clarify the inconsistency in Figure 3 H/I regarding Chmp5 ChIP data normalization. In (H) Chmp5-HA ChIP resulted in normalized values (to IgG) around 5-6, while in (I) JQ1 treatment resulted in the same values.

(e) "Chmp5 deficient cells" is inaccurate for cells subjected to Chmp5 knockdown via shRNA.

(f) Please clarify why there is a second band for CHMP5 (Figure 1B) in T-cell #1, Primary-ALL #P1, #P2, #P3 sample and the same band is absent in rest of the samples.

(g) It would be interesting to add a Vps4 blot for the Figure 1B panel and comment on the fluctuating levels of Vps4 in Extended Data Figure 1B.

(h) We did not understand the following statement: "Reflecting their BRD4-dependency, mRNA levels for T-ALL genes that displayed Pol II stalling (e.g., MYC, XBP1 and TCF7) were comparably downregulated by loss of CHMP5 and by the BET inhibitor JQ1 (Fig. 4I). In fact, despite largely normal (or even higher) BRD4 binding at promoters in these cells (Extended Data Fig. 4A, B), JQ1 had no downregulating effect on BRD4 target genes including XBP1 and TCF7 in CHMP5-deficient T-ALL cells (Fig. 4I)". Please clarify the statement.

Version 1:

Reviewer comments:

Reviewer #1

(Remarks to the Author)

Authors have done a great job at answering my questions (same with other reviewers' questions). I would like to congratulate them once again.

Reviewer #2

(Remarks to the Author)

The authors have satisfactorily addressed each of my critiques. I commend them on a fascinating study.

Reviewer #3

(Remarks to the Author)

The authors have done a great job during the revision and have addressed my major concerns in a adequate manner.

General Comments to All Reviewers

We thank all the reviewers for their careful reading and for insightful feedback on our “very nice” study as recognized by reviewer #1. Importantly, we appreciate their recognition of the novelty of the data which was found to be “compelling” with a “preponderance of evidence” demonstrating transcriptional regulation of T-ALL pathogenesis by the ESCRT protein CHMP5. We agree with the questions and concerns that were raised and have addressed them in this revised manuscript. Of note, we wish to highlight the following key new findings:

1. Whereas CHMP5 deficiency alone did not alter apoptosis, it impaired cell cycle progression, a finding that in part explains the severely defective proliferative capacity of CHMP5-deficient T-ALL cells and their increased sensitivity to chemotherapy (**new Supplementary Fig. 1c-e**).
2. We show that CHMP5 proteins which lack the nuclear localization signal (delta NLS) still interact with the ESCRT partner Vta1 (**new Supplementary Fig. 3e**) but fail to rescue transcription of *MYC* in CHMP5-KD T-ALL cells (**new Supplementary Fig. 4f**). This is in accord with the possibility that CHMP5 might be dispensable for the ESCRT machinery in the T-ALL used in our study as we found no difference in ESCRT-dependent events, including expression of the T cell receptor (TCR), CD4 and CD8 between control and KD cells which both similarly downregulated TCR in response to anti-CD3 stimulation (**new Supplementary Fig. 1f,g**). Nevertheless, in agreement with reviewer #3, we can't exclude that other ESCRT-dependent processes might be altered and impact T-ALL cells.
3. Reinforcing the significance of high CHMP5 expression in T-ALL, we show that “gain-of-function” (overexpression) of CHMP5 not only augmented steady-state T-ALL cell growth but also promoted resistance to chemotherapy. In particular, CHMP5 overexpression increased *MYC* and impaired JQ1-induced cell death, consistent with CHMP5 modulating the BRD4 axis which drives *MYC* expression (**new Fig. 5k to 5m and Supplementary Fig. 6m**).
4. To clarify whether the interaction with BRD4 is required for the stability of p300 proteins (which were upregulated in CHMP5-KD cells in which p300-BRD4 interaction is disrupted), we used a BRD4 PROTAC degrader MZ1 to deplete BRD4 and examined p300 expression. We found that even in CHMP5 sufficient cells, BRD4 degradation, increased p300 protein levels (**new Supplementary Fig. 5d**). The consequence of increased p300 in (CHMP5-deficient) T-ALL cells is presently unclear. Since we found no increase in p300 recruitment at the *MYC* locus for example (**Fig. 4f**), we speculate that p300 might redistribute to other genomic sites to promote transcription of other genes such as the inflammatory program that is upregulated in CHMP5-deficient T-ALL (**Fig. 1a**). Future studies are however required to resolve this issue.
5. Collectively, our findings that CHMP5 nucleates the p300-BRD4 interaction that mediates the epigenetic program required for transcription of key T-ALL genes raised the novel possibility that combined p300-BRD4 targeting would be more efficacious than targeting each cofactor independently in limiting T-ALL. In support of this mechanistic hypothesis, we provide novel *in vitro* evidence that a dual p300-BRD4 inhibitor (NEO2734) currently in clinical trial was more efficacious than single BRD4 (JQ1) or p300 (CCS1477) inhibition (**new Fig. 4j to 4l**).

Overall, by incorporating these new data and reviewer suggestions into the revised manuscript, we believe that our study is now much strengthened and our conclusion that CHMP5 plays a critical role in T-ALL initiation and maintenance reinforced. We propose that T-ALL oncogenes such as activated NOTCH1 hijack CHMP5 to promote the T-ALL program and as a result oncogenes create a “non-oncogene” addiction to CHMP5 that can be exploited in the design of effective T-ALL therapeutics.

Please find below a point-by-point response (**in blue font**) to the raised concerns and suggestions. For clarity, all new data references in this response are highlighted in **yellow**. Furthermore, in the revised manuscript, edited texts are in **red font** and all new or revised figures are highlighted with a **red box**.

Point-by-point Response to Specific Reviewer Comments

Reviewer #1, expertise in T-ALL, NOTCH/MYC, epigenetics and metabolism:

In this interesting article, Umpired-Wilson et al identify CHMP5 as a critical factor in T-ALL. Authors start by showing CHMP5 overexpressed in T-ALL compared to normal cells, and they show the CHMP5 knockdown in T-ALL cell lines sensitizes them to treatments with AraC or Dexamethasone+GSI. Gene expression profiling upon CHMP5 knockdown revealed a clear downregulation of the MYC transcriptional program. Mechanistically, authors show that CHMP5 interacts with BRD4 specifically in the nucleus and show multiple ChIPseq data supporting this interaction mediates PolII pause release and contributes to the formation of the BDME superenhancer. Further, CHMP5 promotes the interaction between BRD4 and p300, leading to increased H3K27 acetylation. Consistently, NOTCH1-induced T-ALL generation was completely abrogated in the absence of CHMP5, and CHMP5-null cells infected with NOTCH1 showed decreased levels of MYC and decreased LIC frequency. I would like to congratulate authors for their very nice study. Still, I have some comments that might further improve the paper (not listed in order of importance):

We thank Reviewer #1 for their compliment and for recognizing the quality of our study. We also thank them for their thoughtful concerns that were raised. We address them below:

1. Authors go on to directly show the effects of CHMP5 knockdown in response to some chemotherapy (the rationale for the use of AraC or Dexa+GSI is somewhat unclear), but they do not show the effects of CHMP5 knockdown on its own, which is relevant, as this would be reflective of effects on T-ALL progression (as opposed to initiation shown in Fig 6). Authors should show proliferation, as well as cell cycle/apoptosis effects of shCHMP5 cells vs shControl cells.

We thank the reviewer for this comment and suggestion. We have now characterized baseline features of CHMP5-deficient T-ALL. While loss of CHMP5 had no impact on baseline apoptosis, CHMP5-deficient T-ALL cells appeared arrested with more cells in S phase and fewer cells in G2/M phase and showed a severe downregulation of cell cycle genes (new Supplementary Figs. 1c-1e, 1j). In line with these results, CHMP5-deficient T-ALL cells (similar to Myc-deficient T-ALL) were impaired in their growth in culture (Fig. 1j) indicating that CHMP5 is required for steady-state T-ALL homeostasis.

We have also clarified in our “proof-of-principle” chemotherapy studies (lines 354-356), that we selected AraC (cytarabine) as a chemotherapy agent because it is currently used as part of an inductive treatment regimen for pediatric and adult T-ALL (PMID: 25966987, 10653870) and mechanisms of resistance to AraC agent that contributes to poor treatment outcomes (PMID: 11054062, 33669053) are not fully understood. Accordingly, our findings that CHMP5-KD cells are more sensitive to Ara-C (Fig. 5e), implicate CHMP5 in Ara-C chemoresistance.

Similarly, we tested Dexamethasone+gamma secretase inhibitors (GSI) as this combination is being evaluated in pre-clinical and clinical trials for activated NOTCH1-driven T-ALL (e.g., CUTLL1 which we used in our study) (PMID: 19098907, 26407235, 33538669). This has been clarified in the revised manuscript (lines 360-370).

The increased sensitivity of CHMP5-deficient T-ALL to these therapeutic agents is likely a reflection of the fundamental requirement for CHMP5 in mediating the T-ALL program.

Ideally, authors could generate a tamoxifen-inducible CHMP5 cKO T-ALL (by breeding the floxed mice with tmx-Cre mice) in order to show the effects of secondarily deleting CHMP5 in already established T-ALL in vivo, but that might be beyond the scope of the present study.

This is a good suggestion! However, as the reviewer acknowledges, given the extensive timeline required for constructing and validating a new genetic mouse model, generating an inducible Cre model for deleting CHMP5 *in vivo* exceeds the scope of the current manuscript. Nevertheless, as our goal in the present study is to provide mechanistic demonstration of the essentiality of the CHMP5-BRD4-p300 axis in T-ALL pathogenesis, we show for the first time a more superior efficacy of dual targeting of BRD4 and p300 (**new Fig. 4j to 4l**).

2. It is unclear how CHMP5 knockdown increases NR3C1 expression. Could authors show any data suggesting a mechanism? if not, this should at least be discussed.

This is an important point raised by the reviewer. In activated NOTCH1 (ICN1)-driven T-ALL such as the CUTLL1 cell line used in our study, NR3C1 expression is repressed by HES1 (PMID:19098907, 22846929) and likely also indirectly by MYC (PMID: 27396325) as depicted (**Rebuttal Fig. 1; Supplementary Fig. 6i**; adapted from PMID: 19098907). Indeed, besides MYC downregulation, we also found that CHMP5-KD T-ALL expressed significantly reduced HES1 (**Supplementary Fig. 6h**), which likely contributed to their upregulation of NR3C1 (mRNA and protein). Moreover, as NR3C1 can also repress NOTCH1 (PMID: 22846929), we suspect that there is a positive feedback loop wherein increased NR3C1 causes a decrease in NOTCH1/ICN1 expression resulting in further downregulation of MYC and HES1 expression. In line with higher expression of NR3C1 in this model (**Rebuttal Fig. 1**), Dexamethasone+GSI treatment was more effective in killing CHMP5-depleted NOTCH1-dependent T-ALL while increasing NR3C1 and BIM targets (**Fig. 5g-5j** and **Supplementary Fig. 6g**). Of note, this was not the case in NOTCH1-independent T-ALL like Loucy in which CHMP5-KD does not perturb NR3C1 (**new Supplementary Fig. 6j to 6l**). We have described these new data and discussed these possibilities in the revised manuscript (**lines 360-370**).

Rebuttal Fig. 1. Model of NR3C1 regulation by CHMP5 in ICN1-driven T-ALL

3. Authors state "CHMP5-regulated mechanisms dictate patient response to chemotherapy" (Line 144). This is a quite strong statement, I would suggest to down tone the "dictate".

We agree and have revised this statement (lines 352-354).

4. In Fig S1B, authors show surprisingly low ICN1 levels for several cell lines, such as DND41, KOPTK1 or JURKAT. Specifically, JURKAT shows the same pattern as LOUCY in the next lane, and Loucy is a NOTCH1-negative cell line. Jurkat should definitely show more ICN1 than Loucy at the very least. I would suggest to repeat this WB and/or verify that all of their cell lines are indeed what they are supposed to be.

Rebuttal Fig. 2. ICN1 (Val1754) antibody validation. Wildtype (B6) DP thymocytes or sorted T-ALL cells from chimera mice generated with ICN1-transduced bone marrow. N = 3 mice/group

We thank the reviewer for this observation. All our cell lines were commercially obtained; hence we considered that this issue might be due to the optimization/lot of our anti-ICN1 antibody combined with the differential expression of ICN1 products depending on the NOTCH1 activation/processing mutation (PMID: 28115368) in the various human T-ALL cell lines. Using newly

acquired and re-validated (Rebuttal Fig. 2) anti-ICN1 antibody (Val1744; Cell Signaling Technology, D3B8; #4147S), we have repeated this western blot and confirmed ICN1 expression in all the T-ALL cell lines except Loucy as expected (new Supplementary Fig. 6b).

5. In Fig 2, authors show correlation of MYC mRNA/protein and CHMP5 protein in their own data. Can they also show this with CHMP5 mRNA? Related to this, it would be more relevant if authors could see this MYC/CHMP5 mRNA correlation in the T-ALL patient dataset from TARGET (Liu et al, Nat Genet, 2017).

We thank the reviewer for these suggestions and agree that showing the correlation of MYC/CHMP5 mRNA in a T-ALL dataset is more clinically relevant. First, in our CHMP5-KD T-ALL cells, we show that MYC mRNA also significantly correlated with CHMP5 mRNA (new Supplementary Fig. 2d), albeit with a lower correlation coefficient compared to correlation with CHMP5 proteins. This is not surprising given that CHMP5 protein levels are more relevant for this phenotype. Importantly, in the TARGET T-ALL dataset, we also found that CHMP5 mRNA also correlated with MYC mRNA expression (new Supplementary Fig. 2e). These analyses are consistent with our mechanistic findings that CHMP5 promotes the transcriptional program in T-ALL that induces MYC expression.

6. Fig 2B and 6F show GSEA of MYC signatures. Even if significant, these plots are not as strong as they could be, which might be due to the fact that these are the MYC TARGET datasets from public databases. I would suggest authors to run their gene expression signature against the specific signature obtained upon deletion of the N-Me enhancer in T-ALL (Herranz et al, Nat Med, 2014), as this dataset is T-ALL specific and might give them a stronger correlation with the T-cell specific MYC-driven program.

We thank the reviewer for this suggestion. In addition to the Hallmark pathway analysis, we have also now assessed differentially expressed genes associated with deletion of the N-Me enhancer (Herranz et al, Nat Med, 2014; PMID: 25194570) in T-ALL to our CHMP5-KD T-ALL dataset. In line with loss of MYC in CHMP5-deficient T-ALL, “UP in N-Me” genes in T-ALL were also upregulated in CHMP5-KD while “DOWN in N-Me” were downregulated in CHMP5-KD T-ALL (new Supplementary Fig. 1h). Additionally, to further and independently evaluate the specificity of MYC-regulated genes, we analyzed a set of genes upregulated or downregulated in MYC-amplified small cell lung cancer (PMID:16116477). In this analysis, whereas genes downregulated by MYC amplification were upregulated in CHMP5-KD T-ALL, genes upregulated by MYC amplification were downregulated in CHMP5-KD T-ALL (new Supplementary Fig. 1i). Collectively, these analyses support that CHMP5-KD cells resemble MYC-deficient/N-Me deficient leukemia and importantly, corroborate that CHMP5 modulates the transcriptional program (exemplified by MYC activity) required for T-ALL homeostasis and pathogenesis.

7. In Fig 3C, cells lacking the NLS seem to be absent from the nucleus. Still, that FLAG WB image is way less exposed than the FL, making this comparison difficult. I would suggest to revise this WB so that the signal in FL and deltaNLS is similar, in order to better assess whether delta NLS is completely absent from the nucleus, or still translocates but to a lesser extent.

We thank the reviewer for this observation. However, we point out that FL and NLS CHMP5 in this WB image are similarly exposed as they were run, transferred, and probed on the same gel and immunoblot. The differential intensity of the NLS-CHMP5 was routinely observed in our experiments and likely arises from the differential expression of this truncation mutant even at the same level of transduction efficiency assessed by GFP expression by which transduced cells were identified and sorted. This effect may be due to the proximity of the NLS (aa33-53) to putative phosphorylation residues (aa26 and aa30) that regulate CHMP5 stability (PMID: 28553951). Quantification across multiple experiments indicated that NLS-CHMP5 less efficiently translocated to the nucleus (new Fig.

2f). Nevertheless, we also now demonstrate in sorted transduced (GFP+) T-ALL cells, that unlike full-length CHMP5-transduced cells, NLS-CHMP5 did not “rescue” expression of T-ALL genes in CHMP5-KD T-ALL cells (new Supplementary Fig. 4f). Together, these data corroborate that the aa33-53 region of CHMP5 likely functions as a putative NLS that promotes its nuclear translocation. Of note, NLS-CHMP5 still interacted with VTA1/Lip5 (new Supplementary Fig. 3e), the canonical ESCRT partner of CHMP5 (PMID:23105106), reinforcing that CHMP5’s function in T-ALL that we uncover in this study is likely due to its transcriptional control function and not the ESCRT machinery.

8. In Fig 4, authors show data supporting pol II pause release. However, they do this by looking at RNA pol II ChIPseq alone. This is better assessed by looking at the ration of pol II phospho-S5(TSS) vs pol II phospho-S2(TE). While performing ChIPseq for these too might be an overkill at this point, authors could perform local ChIP for this specific polII versions at MYC and/or other targets.

This is a valid point as we do not address Pol II phosphorylation status in the paper. Pol II phospho-S5 is assembled at the TSS while Pol II phospho-S2 mediates elongation and thus present at the TES (PMID:19741698, 20434984). As suggested by the reviewer, instead of ChIPseq, we performed ChIP-qPCR at MYC for pS5 and pS2 Pol II to address this. We found that similar to total Pol II, which was enriched at the TSS relative to the TES of the MYC locus in CHMP5-KD cells, there was also more Pol II pS5 at the TSS and less Pol II pS2 at the TES in CHMP5-KD cells (new Supplementary Fig. 4e). Importantly, the ratio of Pol II S5/S2 was significantly higher in CHMP5-KD cells (new Fig. 3j), consistent with promoter proximal stalling of Pol II. These results support that CHMP5 regulates transcription at least in part by controlling Pol II pause release that is governed by the BRD4/p300/MYC axis (PMID:35866356).

9. In Fig 4 it is slightly confusing the nomenclature used by authors when they just show/say "SE". Is this the BDME enhancer? I would suggest to consistently show in the figures both N-Me and BDME and avoid confusions. Related to this, I think it is relevant to show N-Me profile (BRD4, H3K27ac...) everywhere. N-Me is the primary regulator of the NOTCH1-MYC axis in T-cells, so it is important to show potential effects in all of their data, even if in Fig 3 they didn't show BRD4/CHMP5 binding there. I would be surprised if authors observed such drastic effects on MYC without N-Me being more affected (maybe H3K27ac data in Fig 4 and S4?).

We thank the reviewer for this suggestion and have now rectified the nomenclature across the manuscript figures and texts to NDME (N-Me) and BDME.

As suggested, we now also show BRD4 and H3K27Ac tracks for the NDME as well. While BRD4 and H3K27Ac peaks were also reduced in the NDME, these enrichment changes were not as profound as the H3K27Ac in the BDME (revised Fig. 3n). This data suggest that CHMP5 has more of an impact on the BRD4-dependent MYC enhancer (BDME) than on the NOTCH-mediated MYC (NDME). Why this is the case is presently unclear and may be due to our observation that CHMP5 does not interact with ICN1 (Supplementary Fig. 3f) but instead with BRD4 (p300) which acetylates the BDME.

Related to this, N-Me loss has a strong effect on normal T-cell development (and MYC expression throughout it) but, according to authors, CHMP5 effects are milder. Authors state that CHMP5-null DP cells lack MYC (line 479) but i couldn't find this data here or on their Nat Immunol paper. Can authors show this? Maybe taking advantage of their MYC-GFP tag?

We think the reviewer might be referring to this line: “Intriguingly, CHMP5 is dispensable for normal CD4⁺CD8⁺ thymocyte generation (that lack MYC) but required for MYC⁺CD4⁺CD8⁺ T-ALL cells”. When we stated that they lack MYC, we meant that at baseline, DP thymocytes (with or without CHMP5) do not express MYC and are not impacted by loss of CHMP5, data that we now show using

the MYC-GFP reporter (**new Supplementary Fig. 7j and 7k**). By contrast, oncogenic ICN1-induced DP T-ALL cells highly express MYC-GFP but such MYC-GFP induction is abrogated in CHMP5-deficient (i.e., Cd4-Cre⁺Chmp5^{fl/fl}) ICN1+ thymocytes (**new Supplementary Fig. 7j and 7k**). These findings suggest that activated NOTCH1 signals induce MYC expression by a mechanism dependent on CHMP5. In further test of this notion, we stimulated WT or CHMP5-deficient primary thymocytes with plate bound NOTCH1 ligand DLL4-Fc (PMID: 21700774) and found that CHMP5-deficiency similarly impaired MYC induction (**new Supplementary Fig. 7l**). Thus, CHMP5 is required for the activated NOTCH1 (ICN1)-induced transcriptional program exemplified by MYC.

Unlike the N-Me^{-/-} mice in which N-Me deletion is germline and severely depleted DN/ISP precursors of DP thymocytes and MYC expression in DN2/3 thymocytes (PMID: 25194570), our murine *Chmp5*-deficient model using the Cd4-Cre (i.e., Cd4-Cre⁺Chmp5^{fl/fl}) initiates in DP thymocytes and had no impact on DP thymocyte generation and DP thymocyte numbers (PMID:28553951). While outside the scope of the current manuscript, it is possible that deletion of *Chmp5* earlier in DN thymocytes might result in similar severe consequences in DP thymocytes as N-Me.

We have included these new data (**lines 419-426**) and discussion (**lines 459-469**) in the revised manuscript.

10. Authors fail to generate NOTCH1-induced leukemias, and authors conclude this is likely driven by lack of MYC expression. Can authors generate leukemias by overexpressing MYC in CHMP5-null progenitor cells?

This is an interesting question raised by the reviewer. However, in line with previous studies cited below, we were not successful in generating leukemias by retrovirally induced MYC expression alone in hematopoietic stem and progenitor cells (HSPCs). MYC overexpression alone in bone marrow HSPCs resulted in apoptosis and largely yielded myeloid leukemias, although this approach can generate mixed (myeloid and lymphoid) leukemias when co-expressed with Bcl-2 (PMID:15972450, 26308666). While thymocyte overexpression of MYC variably generated thymic lymphomas, the strategy required generation of new transgenic mice (PMID: 8473043; 8617934) given the technical difficulty of transducing primary murine thymocyte subsets. We consider this beyond the scope of the present study.

Importantly, we wish to highlight that our conclusion from the present study is that CHMP5 is required to promote the BRD4-p300-dependent T-ALL program driven by oncogenes like ICN1 and that downregulation of MYC (a BRD4 target gene) exemplifies this pathway. As the BRD4-p300 axis regulates a plethora of genes (e.g., cell cycle genes (**new Supplementary Fig. 1j**) and metabolic genes (**new Supplementary Fig. 2g**), we cannot exclude that other genes apart from MYC likely also contribute to the phenotype of CHMP5-deficient T-ALL.

=====

Reviewer #2, expertise in leukemia epigenetics:

In this study, Umphred-Wilson et al. demonstrate an intriguing novel nuclear function for the protein CHMP5. CHMP5 has been characterized as a cytoplasmic factor involved in endosomal sorting and membrane remodeling (ESCRT machinery). Yet here the authors demonstrate with a wealth of data that CHMP5 has a nuclear localization sequence, can transport to the nucleus where it can associate with chromatin together with BRD4 and p300 to promote gene transcription. They also demonstrate in cell lines and primary patient samples that CHMP5 is upregulated in T-cell acute lymphoblastic leukemia compared to normal T cells, and so interaction with BRD4-p300 at enhancer elements contributes to MYC expression. Genetic knockdown of CHMP5 decreases the MYC expression

program in T-ALL and can decrease T-ALL cell proliferation both in vitro and in vivo. The preponderance of data included in the paper convincingly demonstrates CHMP5 involvement in transcriptional outputs in T-ALL. I have several minor questions that should be addressed:

We thank Reviewer #2 for their careful review of our study and their feedback (which we address below) that has helped improved the revise manuscript. As acknowledged by the reviewer, we believe that “wealth of data” supports the critical dependency of T-ALL on CHMP5.

1) The K-M data in Figure 1D and Ext. Data Fig 1D seem to be replicated, although it purports to show different cohorts defined by expression of either CHMP5 or a control gene.

We thank the reviewer for this observation. This was an error in assembling the manuscript figures. As shown in the now revised manuscript, Fig. 1D (**revised Fig. 5d**) shows the Kaplan-Meier survival data of TARGET T-ALL patients stratified by levels of *CHMP5* mRNA expression while Ext Data Fig. 1D (**revised Supplementary Fig. 6d**) shows Kaplan-Meier survival data of the same patients stratified by *VPS4A* mRNA expression. Notably, high or low CHMP5 correlated with poor and improved prognosis respectively ($P = 0.017$), whereas high and low *VPS4A* patients were not distinguishable ($P = 0.86$).

2) It is interesting that CHMP5 KD can have a significant anti-MYC effect, with dramatically reduced cell proliferation (Fig 2J), and yet there is no observable apoptosis and very minimal cell cycle arrest phenotype. Are other antiproliferative mechanisms a consequence of CHMP5 loss?

We thank the reviewer for this question. We were also surprised by the severe defect in cell proliferation in CHMP5-KD T-ALL despite no overt increases in apoptosis which we additionally confirmed with Caspase-3 staining (**new Supplementary Fig. 1c**). In addition to their downregulation of several cell cycle genes (**new Supplementary Fig. 1j**), another factor that likely also contributed to the severe proliferative defect is the significant downregulation of multiple energy metabolic pathway genes which consequently resulted in >80% decline in glycolysis and oxidative phosphorylation capacity of CHMP5-KD T-ALL cells at baseline (**revised Fig 1k and 1l**).

3) Does overexpression of CHMP5 in normal T cells or CHMP5-low T-ALL induce BRD4 activity and MYC expression?

We thank the reviewer for this question. However, we note that overexpression of CHMP5 in normal T cells presents with potential confounding issues. Specifically, anti-CD3/CD28 pre-activation of T cells to enable retroviral or lentiviral transduction not only induces 3-6-fold more CHMP5 in normal T cells (our observation, manuscript in preparation), but also considerably induces Myc (PMID:22195744; our observation). Hence, the outcome of overexpressing CHMP5 in normal T cells would be ambiguous. Indeed, our preliminary attempts did not yield any appreciable overexpression of CHMP5 in normal (activated) T cells.

Nevertheless, we investigated the consequence of overexpressing CHMP5 in two activated NOTCH1 human T-ALL cells KOPTK1 and MOLT-3 that were CHMP5 low (compared to CUTLL1). Remarkably, overexpression of CHMP5 increased proliferation of these T-ALL (**new Fig. 5k**). Overexpression of CHMP5 also limited the effect of BRD4 inhibition by JQ1 (**new Fig. 5m**) while increasing MYC (**new Supplementary Fig. 6m**). These results are in line with our findings that CHMP5 promotes BRD4 driven T-ALL transcription program. Interestingly, CHMP5 overexpression also limited the suppressive effect of the T-ALL chemotherapy vincristine (**new Fig. 5l**), consistent with our TARGET T-ALL patient survival data in which high expression of CHMP5 correlated with worse survival (**Fig. 5d**).

4) The authors claim that BRD4 signal is specifically lost at distal enhancers upon CHMP5 loss, and demonstrate this at selected loci - can they show this cumulatively across the epigenome?

We thank the reviewer for this question. We now show BRD4 signals across the genome using a “hockey stick” plot (new Supplementary Fig. 4k). While there is a broad disruption BRD4 across loci, it was less striking than changes in H3K27ac depicted in a hockey plot (Fig. 3m), likely because H3K27 acetylation is also impacted by the loss of p300 which is also disrupted in CHMP5-KD T-ALL (Fig. 4).

5) There is a striking upregulation of p300 expression upon CHMP5 loss. Can the authors speculate as to the consequences of this? p300 is known to 'redistribute' to other regulatory sites in the epigenome upon inhibition - where does all the newly expressed p300 go?

We appreciate this question by the reviewer. However, the basis of this upregulation and the consequence of p300 upregulation is presently unclear. We suspect that this increase in p300 is due to its diminished interaction with BRD4 whose kinase activity is known to target proteins (e.g. MYC) for degradation (PMID:32482868). Indeed, using a BRD4 PROTAC MZ1 we found that loss of BRD4 also increased p300 expression in CHMP5-sufficient T-ALL cells, supporting that interaction with BRD4 likely controls p300 protein homeostasis in T-ALL cells (new Supplementary Fig. 5d).

As the reviewer suggests and as previously reported including in cells in which p300 is inhibited (PMID: 34019788; 25051172), p300 proteins in CHMP5-depleted T-ALL cells might “redistribute” to other genomic sites where they cooperate with other factors to mediate expression of other genes, potentially including genes that were upregulated in CHMP5-KD T-ALL cells. For example, prostate cancer cells in which p300 is inhibited upregulate multiple inflammatory genes (PMID: 33431496) similar to CHMP5-KD T-ALL (Fig. 1a). Additionally, in p300 overexpression in fibroblast promoted senescence genes and impaired cell growth (PMID: 30773298) in line with the impaired growth of CHMP5-KD T-ALL cells. As we expound in our discussion, definitive resolution of the basis and consequences of upregulated p300 in CHMP5-KD T-ALL will require future work beyond the scope of the present study (lines 510-520).

6) The identity of the loading controls used in Ext Data Figure 3A is missing.

Thank you for this observation. We have now added loading controls (Lamin or vinculin) and CHMP5 labels as well as molecular weight markers to the revised figure.

=====

Reviewer #3, expertise in endosomal sorting complex required for transport:

The manuscript by Katharine Umphred-Wilson and colleagues presents compelling evidence implicating Chmp5, an ESCRT-III subunit, in the promotion of T-ALL. Through a comprehensive series of experiments conducted in both cellular and murine models, the authors propose a mechanism wherein a nuclear fraction of Chmp5 interacts with BRD4 and p300, to augment their recruitment to super-enhancers and subsequently upregulating the transcription of various T-ALL-associated genes, including MYC. The loss of Chmp5 leads to a significant decrease in MYC expression and other T-ALL genes, rendering Chmp5-depleted cells more susceptible to chemotherapy. While the findings are intriguing and the experiments well-executed, there are two critical points that require clarification to strengthen the conclusion that exclusively nuclear Chmp5, in conjunction with BRD4 and p300, governs the transcriptional effects on T-ALL genes and hence T-ALL pathogenicity.

We thank the reviewer for their expert review of our study and for emphasizing the need to distinguish the intersection between the ESCRT and chromatin/transcription regulation function of CHMP5 in T-ALL pathogenesis. New data addressing reviewer concerns (as described below) reinforce that CHMP5 regulates p300-BRD4-dependent transcription in the nucleus by a mechanism that is in part dependent on the NLS of CHMP5. Notably, NLS-CHMP5 is still able to interact with VTA/LIP5 in the ESCRT machinery (new Supplementary Fig. 3e) but unable to rescue gene expression in CHMP5-deficient T-ALL cells (new Supplementary Fig. 4f). Nevertheless, given that T-ALL and other cancer cells utilize ESCRT for other cellular processes, we cannot completely exclude that defects in the ESCRT machinery might also contribute to the CHMP5-deficiency phenotype in vitro and/or in vivo. As suggested by the reviewer, we have included this limitation in our discussion (lines 470-483).

Major Points:

(1) While the majority of experiments demonstrate the necessity of Chmp5 for T-ALL gene expression, it remains unclear whether solely nuclear Chmp5, in concert with BRD4 and p300, accounts for the observed effects. To better discern nuclear from cytosolic Chmp5 functions, such as the down-regulation of Notch signaling by ESCRTs or the stabilization of the pro-survival protein Bcl-2, we recommend generating point mutations in the nuclear localization signal (NLS) of Chmp5, preserving cytosolic function while impeding nuclear import. Alternatively, identifying mutations in Chmp5 that specifically disrupt its binding to BRD4 or p300 would be helpful. Expressing these mutant Chmp5 variants in Chmp5-deficient cells could elucidate the distinct roles of nuclear and cytosolic Chmp5. If these experiments exceed the study's current scope, the authors should tone down their conclusions and entertain the possibility of non-nuclear Chmp5 involvement in T-ALL gene expression.

We appreciate these insightful suggestions. We have now used a number of approaches to attempt to distinguish the ESCRT versus non-ESCRT contributions of CHMP5 to T-ALL pathogenesis. First, we biochemically assessed CHMP5 interaction with chromatin in soluble and chromatin nuclear fractions and found that nuclear CHMP5 associated only with chromatin and was not present in the soluble nuclear fraction (new Fig. 2k), as would be predicted if CHMP5 were part of an ESCRT complex in the soluble nuclear fraction.

Next, given that ESCRT proteins mediate membrane receptor recycling/endocytosis and cytokinesis, we compared basal expression levels of the T cell receptor (TCR), CD4 and CD8 coreceptors on control and KD cells and found that these were largely indistinguishable (new Supplementary Fig. 1f). Moreover, TCR downregulation by anti-CD3 activation was comparable in control and CHMP5-depleted T-ALL (new Supplementary Fig. 1g). Finally, cell size (FSC; Supplementary Fig. 2h) and DNA content (indicated by 7-AAD on x-axis; Supplementary Fig. 1e) were not increased in CHMP5-KD T-ALL cells as would be expected in ESCRT-deficient cells (PMID:20616062).

Furthermore, as point mutants that abrogate CHMP5 binding to BRD4 or p300 or abrogate its nuclear localization are currently unknown, we used our CHMP5 mutant (Fig. 2d) to test the impact of nuclear CHMP5 on T-ALL gene transcription. Notably, like full-length (FL) CHMP5, the deltaNLS CHMP5 still interacted with VTA1/Lip5, the canonical ESCRT partner of CHMP5 (PMID: 15644320, 23105106) (new Supplementary Fig 3e). However, whereas full-length CHMP5 “rescued” expression of T-ALL genes like MYC and XBP1 (Fig. 1f-g and new Supplementary Fig. 4f), delta-NLS CHMP5-transduced CHMP5-KD T-ALL cells still expressed significantly lower amounts of MYC and XBP1 compared to control (CT) T-ALL cells (new Supplementary Fig. 4f).

Collectively, these results support that nuclear-imported CHMP5 is bound to chromatin, at least in part through its interaction with BRD4, and promotes BRD4-p300 driven transcription. As Bcl-2 proteins were not altered in CHMP5-KD T-ALL cells (revised Fig. 1c), we consider the nuclear role of CHMP5 in T-ALL to be mechanistically distinct from its posttranslational role in normal T cell development in which CHMP5 promoted Bcl-2 protein stability (PMID: 28553951). It is noteworthy that common to its transcriptional regulation function in T-ALL and posttranslational control of client proteins in normal thymocytes, is that CHMP5 functioned as an “adaptor” that promoted protein-protein interaction.

Nevertheless, we agree that our findings do not formally exclude that CHMP5's function in the ESCRT machinery might also contribute to its requirement for T-ALL pathogenesis. As additional experiments, beyond the scope of the current study, will be required to formally discriminate ESCRT vs non-ESCRT CHMP5 contribution to T-ALL, we have "toned" down our conclusions in the discussions (**lines 470-483**).

(2) In prior research, the authors convincingly demonstrated Chmp5's necessity for T cell development, partially through Bcl-2 stabilization, with Chmp5 loss resulting in impaired T-cell development. Given these findings, it is perhaps unsurprising that mice lacking Chmp5 in T cells fail to develop T-ALL. Please explain how Chmp5's posttranslational role in T-cell development relates to its nuclear function in T-ALL development.

We thank the reviewer for highlighting our previous studies where we demonstrated the requirement for CHMP5 in T cell development that was in part due to CHMP5-dependent stabilization of Bcl-2 (PMID: 28553951). Interestingly, Bcl-2 proteins were not altered in CHMP5-KD T-ALL (Bcl-2 data has been added to **revised Fig. 1c**), underscoring that the function of CHMP5 in T-ALL is mechanistically distinct from its posttranslational role in normal T cell development where it promoted Bcl-2 stability.

Furthermore, while phenotypically both CD4⁺CD8⁺ (double positive, DP) normal DP thymocytes and activated human or murine NOTCH1 (ICN1)-driven T-ALL display distinct dependency on CHMP5. In our previous study, we specifically mapped dependency on CHMP5 to CD4⁺CD8^{lo}CD69⁺ (TCR-signaled) post-selection stage of thymocyte development and showed that normal (pre-selection) DP thymocytes were in fact not dependent on CHMP5 (PMID: 28553951). In contrast, in the current study, we show that ICN1-induced DP thymocytes are dependent on CHMP5 (**new Supplementary Fig. 7j**). Our data from the present manuscript suggest that this CHMP5-dependency likely arises because of oncogenic ICN1-imposed dependency on BRD4 which upregulates MYC in these T-ALL subtype (PMID:23791182, 24584072, 28115368). Accordingly, in contrast to the high levels of MYC (detected by a MYC-GFP reporter) in WT ICN1+ T-ALL chimera, Chmp5-KO donor ICN1+ cells were severely depleted of MYC (**new Supplementary Fig. 7k**). Further corroborating that NOTCH1 signals impose CHMP5 dependency in DP thymocytes, stimulation with recombinant delta-like 4 (DLL4)-Fc NOTCH1 ligand induced MYC in WT but not CHMP5-deficient (*Chmp5^{fl/fl}Cd4-Cre⁺*) DP thymocytes (**new Supplementary Fig. 7k**). Moreover, Chmp5-KO donor ICN1+ cell transcriptome was enriched in genes that were upregulated in "N-Me" T-ALL cells (**new Supplementary Fig. 7o**) in which MYC expression is impaired by genetic inactivation of the NOTCH1(ICN1)-specific ("N-Me") enhancer (PMID: 25194570).

Minor Points:

(a) Please denote "Mr" for Western blotting.

We have now included the molecular weights for all western blots.

(b) Chmp5 localization should be assessed via immunofluorescence microscopy.

We thank the reviewer for this suggestion and have now performed confocal immunofluorescence (IF) microscopy in both CUTLL1 and a PDX T-ALL. Importantly, we validated the specificity of CHMP5 antibody staining which showed significantly reduced CHMP5 staining in CHMP5-KD T-ALL cells (**new Supplementary Fig. 3b**). Notably, confocal IF also revealed nuclear CHMP5 staining in CUTLL1 and a PDX T-ALL sample (**new Fig. 2b,c**). Thus, both imaging and protein western blot confirm CHMP5 localization in the nucleus in T-ALL cells.

(c) Western blot quantifications (e.g., Figure 3A) do not seem to align with the blots; it seems unlikely that 20% of Chmp5 in SUPT1 cells are nuclear.

Thank you for this observation. All western blot bands were quantified using ImageJ software in which bands were defined by rectangular boundaries. Re-quantification of this band indicated a ratio of ~0.18. Overall, by western blot of cell fractions and immunofluorescence microscopy, we detected CHMP5 in the nucleus of T-ALL cells. Western blot quantification indicated that between 10-50% of CHMP5 localized to the nucleus across multiple T-ALL subtypes represented by PDX T-ALL, CUTLL1, SUPT1, and MOLT3, the latter which was newly included in the revised manuscript (**revised Fig. 2a**).

(d) Please Clarify the inconsistency in Figure 3 H/I regarding Chmp5 ChIP data normalization. In (H) Chmp5-HA ChIP resulted in normalized values (to IgG) around 5-6, while in (I) JQ1 treatment resulted in the same values.

Thank you for bringing this to our attention. In Figure 3H/I (**now revised Fig. 2m/n**) the values were likely influenced by the overall levels of CHMP5-HA expression achieved as both were independently performed and normalized to isotype (IgG) within each experiment. This trend was reproducible across multiple experiments. We nevertheless have repeated this experiment in transduced cells with higher expression of CHMP5-HA and this yielded CHMP5 enrichment across the *MYC* locus that are comparable to that in experiments in which cells were treated with JQ1 (**revised Fig. 2n**).

(e) "Chmp5 deficient cells" is inaccurate for cells subjected to Chmp5 knockdown via shRNA.

We agree and have corrected this to "depleted" or "knockdown" where applicable. Thank you.

f) Please clarify why there is a second band for CHMP5 (Figure 1B) in T-cell #1, Primary-ALL #P1, #P2, #P3 sample and the same band is absent in rest of the samples.

We thank the reviewer for this keen observation. This second band likely corresponds to an alternatively spliced isoform of human CHMP5: the major isoform of CHMP5 (UniProt: Q9NZZ3-1) is 219 amino acids (aa) long and the smaller CHMP5 isoform is 171 aa (UniProt: Q9NZZ3-2). The small isoform lacks the c-terminal 166-219 aa residues. Depending on T-ALL cell type or duration of protein blot exposure, both isoforms can be detected by the polyclonal anti-CHMP5 antibody (Thermo Fisher Scientific, #PA5-63303) used in our study which is raised against aa residues 118-168 that is present in both isoforms. How expression of the smaller CHMP5 isoform (or whether it has distinct functions) is regulated is unclear. We note that our work on CHMP5, including the present study, is focused on the major (219 aa) isoform which migrates at ~35 kDa and is 99% homologous to CHMP5 in mice which only have the 219aa isoform. We have indicated this in the **Methods** section (**lines 719-722**).

(g) It would be interesting to add a Vps4 blot for the Figure 1B panel and comment on the fluctuating levels of Vps4 in Extended Data Figure 1B.

We agree about showing Vps4 in Figure 1B (**now Fig. 5b**) and have re-probed this membrane with a VPS4A antibody. Notably, and similar to another ESCRT protein CHMP1A, VPS4A expression was comparable in PDX T-ALL samples compared to normal T cells (**revised Fig. 5b**).

We are unsure of the reason for the fluctuating levels of VPS4A between T-ALL cell lines Extended Data Figure 1B (**revised Supplementary Fig. 6b**). These may be either due to the mutational status or proliferative dynamics of these cells. Nevertheless, in response to Reviewer #1, we have repeated

this western blot using CHMP1A as our ESCRT control and confirmed that compared to normal T cells, human T-ALL cells express higher CHMP5 proteins whereas CHMP1A proteins were similar between T-ALL and normal T-cells (**revised Supplementary Fig. 6b,c**).

The higher expression of CHMP5 in T-ALL and not ESCRT proteins like VPS4A and CHMP1A is in line with the lack of correlation between VPS4 or CHMP1A expression levels with patient survival in the TARGET T-ALL cohort (**revised Supplementary Fig. 6d,e**).

(h) We did not understand the following statement: "Reflecting their BRD4-dependency, mRNA levels for T-ALL genes that displayed Pol II stalling (e.g., MYC, XBP1 and TCF7) were comparably downregulated by loss of CHMP5 and by the BET inhibitor JQ1 (Fig. 4I). In fact, despite largely normal (or even higher) BRD4 binding at promoters in these cells (Extended Data Fig. 4A, B), JQ1 had no downregulating effect on BRD4 target genes including XBP1 and TCF7 in CHMP5-deficient T-ALL cells (Fig. 4I)". Please clarify the statement.

Thank you for pointing out the lack of clarity in this statement in reference to Fig. 4I (**now Fig. 3i**) and Extended Data Fig. 4A,B. We have now revised it (**lines 231-235**) to better convey our meaning that genes whose transcription is impaired (i.e., high Pol II traveling ratio) by CHMP5 depletion (e.g., MYC, XBP1, TCF7, **now Fig. 3g**) are indeed BRD4 dependent. Hence, these genes were downregulated in CHMP5-KD cells to similar degrees achieved by the BRD4 inhibitor JQ1 (**now Fig. 3i**). Of note, MYC expression was further downregulated in CHMP5-KD cells (**now Fig. 3i**). This suggests that CHMP5-independent but BRD4-dependent mechanism also likely contribute to driving MYC expression in T-ALL.